# Buddhist Ritual and the Bronze Buddha Mold Excavated from the Western Five-Story Stone Pagoda of Hwaŏm Temple, Korea

**Young-ae Lim**

Department of Art History, Dongguk University, Seoul 04620, Korea; yalim0105@dongguk.edu

**Abstract:** This paper examines the bronze Buddha mold that was excavated from the western pagoda of Hwaŏm temple 華嚴寺. The research centers on the mold's date of production, its function, and the reason it was enshrined in the Hwaŏmsa pagoda. The pagoda itself was constructed in the ninth century and is considered to be a Dharani pagoda because *Wugoujingguang datuoluonijing* (無垢淨光大陀羅尼經, *The Great Dharani Sūtra*) is enshrined within the structure. The act of placing the Buddhist scriptures in the pagoda was to benefit the structure's benefactors by absolving them of their sins and granting blessings in their afterlives for their meritorious deeds. Of all the dhāraṇī, *Wugoujingguang datuoluonijing* is the most detailed and particularly emphasizes the act of repetition. The clarity and simplicity of its instructions made it especially popular in eighth-to-ninth-century Korea. The Hwaŏmsa Buddha mold was one of the tools used in the ritual described by *Wugoujingguang datuoluonijing*. Considering the sūtra's insistence on repetition and replication, the mold was a very suitable implement. The use of inexpensive clay also allowed for the mass production of Buddha images that any individual could commission at little cost. Furthermore, this method of producing Buddha images made it easy for the temple to attract followers and thus raise funding for the construction of the pagoda. The clay Buddhas themselves were small, making it possible for one to keep the image on his person and carry it wherever he went. Ultimately, these actions were meant to bring the individuals closer to Buddha and his world.

**Keywords:** Hwaŏmsa; Hwaŏm temple; Hwaŏmsa western stone pagoda; bronze Buddha mold; clay Buddha; *Wugoujingguang datuoluonijing*; *The Great Dharani Sūtra*; Unified Silla 統一新羅 (668–935); dhāraṇī ritual

---

## 1. Introduction

In 1995, a bronze mold was excavated from the western five-story stone pagoda of Hwaŏm temple 華嚴寺 located in Chirisan mountain 智異山 of Chŏllanamdo 全羅南道, South Korea.[1] The bronze mold was an implement that was used in the production of Buddha images. A handle affixed to the mold's rear allowed the user to employ it as a stamp. The side opposite features an incised design of a Buddha figure and two pagodas (Figures 1 and 2). Central to the design is a seated Buddha holding his hands in Dharmacakra-mudrā 轉法輪印 (turning the Wheel of Dharma, a teaching gesture) who is

---

[1]  Hwaŏm temple was erected based on the ideological foundation of the Silla period Avatamsaka sect 華嚴宗 of Buddhism. Though a lack of extant historical records make it difficult to determine an exact date of construction for Hwaŏm temple, scholars have come to the consensus of the mid-eighth century. The western five-story stone pagoda of Hwaŏm temple was previously designated as Treasure no. 133 in 1995. In the same year, the pagoda underwent deconstruction, and artifacts were discovered in the base and body portion of the pagoda. Subsequently, all artifacts discovered in the pagoda were collectively appointed as Treasure No. 1348.

flanked on either side by a pagoda. The mold itself resembles a conventional stamp intended for use on paper. However, a closer inspection reveals a marked depth to the incision of the design that suggests the mold was purposed for an alternative medium. The high relief nature of the mold's design would have been quite suitable to impress materials such as clay, which is of greater firmness than paper. The resulting design was produced in mass quantities, and molded images of clay Buddhas have been found in abundance across East Asia. However, though clay Buddhas are rather common, the same cannot be said for bronze molds. In fact, the example that was excavated from Hwaŏmsa was the first of its kind to be discovered as well as the earliest dated.[2]

The Hwaŏmsa bronze Buddha mold is valuable in that it is rare but also noteworthy because of its findspot. The mold was excavated from the base of the pagoda structure rather than the body, where objects are typically enshrined.[3] Such is the case for the two Buddha statues found in the three-story stone pagoda of the Hwangbok temple site 皇福寺址 and the single Buddha statue discovered in the five-story stone pagoda of Nawŏn-ri 羅原里.[4] All three Buddha images were stored in the body of the pagoda and enclosed inside a sarira reliquary. Likewise, the Hwaŏmsa pagoda also contained a sarira reliquary and various other objects including a sūtra enshrined in the first story of the pagoda's body. Yet amongst the enshrined objects, Buddha images are conspicuously absent from the Hwaŏmsa pagoda. Thus, the placement of the bronze Buddha mold in the pagoda's base and the lack of means to store the mold in forms such as a reliquary render the object a significant find.[5]

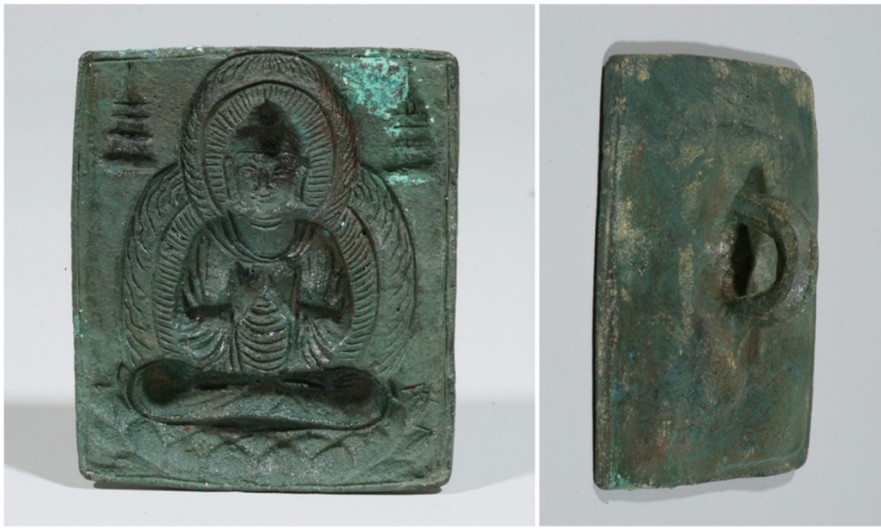

**Figure 1.** Bronze Buddha mold (**left**) front of mold (**right**) back of mold, Unified Silla, late seventh-to-eighth century, 7.1 × 8.2 cm. Western pagoda of Hwaŏm temple, Hwaŏmsa Buddhist Museum [©CBM].

---

[2]  There are other instances of bronze molds such as the example cautiously dated to the Late Paekje period that was excavated at the site of Chŏllabukto 全羅北道 Kimje 金堤. However, the design has been set in relief rather than incised, indicating that the mold was meant for use in mass production. For research on the Kimje plaques, see (Hwang 1980, pp. 3–9).

[3]  When objects of offering, including sarira, are enshrined in a pagoda, they are most often placed in the first, second, or third story of the structure's body. Though there is no specific pattern dictating the placement of these objects, they are usually found in the second or third story of seventh and eighth century pagodas and the first story of ninth century pagodas.

[4]  Both Hwangboksa and Nawŏn-ri were temples located in Kyŏngju. At present, the sites are plots of vacant land with the exception of the surviving pagodas of each temple.

[5]  Though excavations of the pagoda were carried out in 1995, an excavation report has not yet been published. Hence, there are no photographic records of the site at the time of discovery.

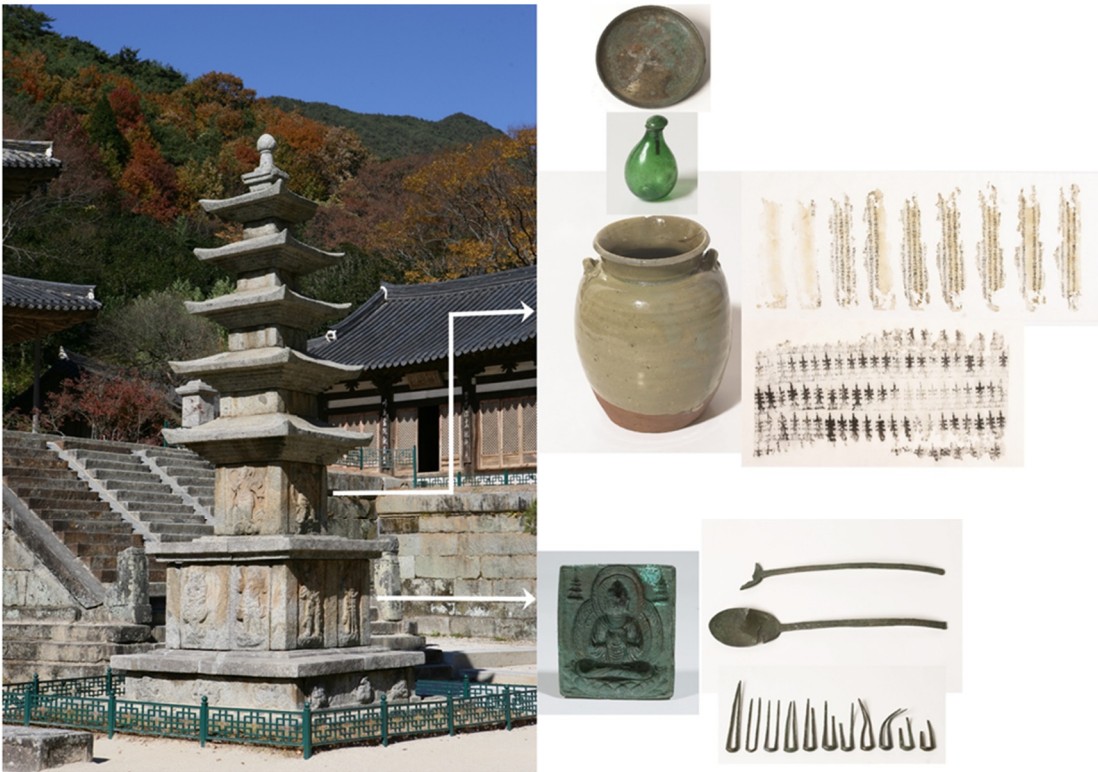

**Figure 2.** Hwaŏmsa western stone pagoda and objects excavated [©CBM, Lim edit].

There are, of course, examples of Buddha statues excavated from the bases of pagodas. There are about ten such examples in the area of what was formerly Kyŏngju 慶州, the capital of Unified Silla 統一新羅 (618–935). These statues are all relatively small, usually no taller than five centimeters and excavated from the bases of various pagodas such as the eastern three-story stone pagodas of the Kamūn temple site 感恩寺址 and Pulguk temple 佛國寺 (Figure 3).[6] It was clearly not an atypical practice to enshrine Buddha images at the base of pagoda structures. However, the Hwaŏmsa find is unique in that the enshrined object is not a Buddha statue but rather a mold for the purposes of producing the Buddha images. Furthermore, the mold was discovered without a protective exterior and among familiar objects including two spoons and eleven hairpins. To such a degree, one of the objectives of this paper was to study the circumstances of acquiring a bronze mold only to place it in the base of a pagoda among everyday items without a covering.

There has always been scholarly interest in the Hwaŏmsa bronze Buddha mold.[7] For the most part, research on the subject has remained focused on the production period of the mold itself or on an analysis of the image's design. This paper is a different approach that assesses the bronze mold in the context of its discovery site, assesses its function, and attempts to clarify the decision to enshrine the mold at the base of Hwaŏmsa pagoda.

---

[6] There is an abundance of small gilt-bronze Buddha statues among the larger temples. These include two in the upper base of the eastern three-story pagoda of Kamūnsaji, one in the upper base of the three-story stone pagoda of Pulguk temple, and one in the base of the three-story stone pagoda of the Mit'an temple site 味寺址. (Kim 2014, pp. 9–10). For research on the Buddha images enshrined in the pagodas, see (I 2016, pp. 56–310; Kim 2016, pp. 40–69).

[7] The most representative scholarly research on the subject of the bronze Buddha mold is a paper by Professor Ch'oe Sŏngŭn (Ch'oe 2000, pp. 24–46). This paper focused on dating the bronze mold to the late ninth century or the early tenth century. This conclusion was based on the fact that the image of a seated Buddha among twin pagodas was popular in the Pala sculpture of India as well as on the connection to the molded clay bricks of Tang China. However, the actual nature of the mold's utility, which is the focus of this paper, was not explored in Professor Ch'oe's research. Likewise, though the Hwaŏmsa mold has been referenced in several publications by scholars such as Kim Lena and Pae Chintal, it was not their primary research focus (Kim 2003a, pp. 228–32; 2007, pp. 68–70; Pae 2003, pp. 220–29; Ch'oe 2004, p. 53).

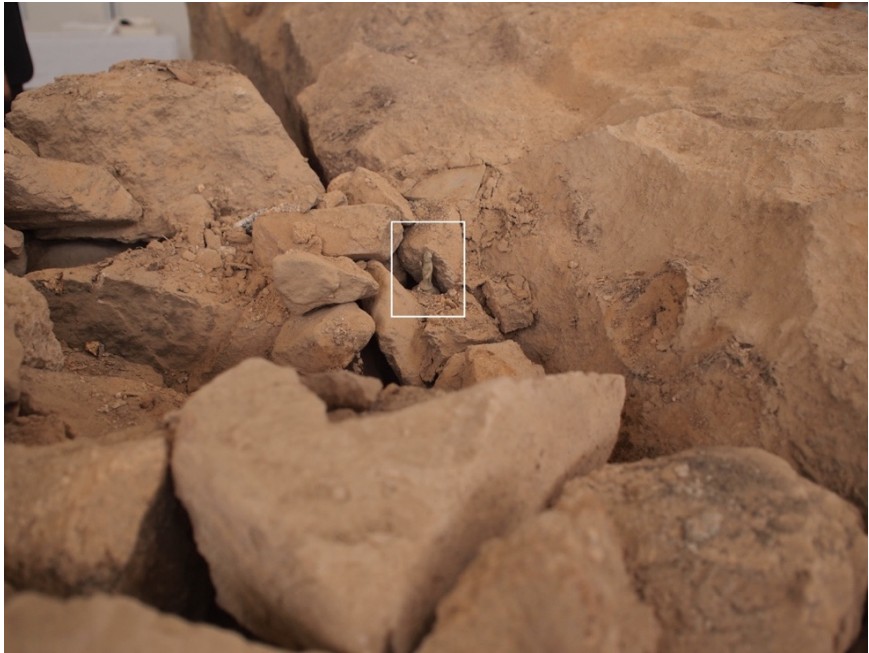

**Figure 3.** Gilt-bronze standing Buddha, Unified Silla, eighth century, H.4.8 cm. Three story stone pagoda of Pulguk temple, Pulguksa Buddhist Museum [©Photograph by Kim Tongha, 2013].

## 2. The Western Five-Story Stone Pagoda of Hwaŏm Temple and the Excavated Buddhist Artifacts

Considered one of the representative temples of Korea, Hwaŏm temple in Chirisan of Chŏllanamdo was constructed in the eighth century by the Buddhist priest Yŏn'gi 緣起 (fl. mid-eighth century).[8] In front of the temple's main building, two pagodas (Figures 4 and 5) were constructed in the ninth century during the Unified Silla period. The bronze Buddha mold was discovered at the base of the western pagoda. Though the two pagodas may seem to be a pair, it is inaccurate to identify them as twin pagodas that were constructed simultaneously.[9] First, although the two pagodas are located across the main building, one each to the structure's left and right, they are not situated symmetrically to one another.

Moreover, the form of the two pagodas and the excavated relics in each structure present significant differences.[10] In form, the eastern pagoda has a one-story base, whereas the western pagoda's base consists of two stories. The eastern pagoda's surface is undecorated, but the opposite is true for its western counterpart. The western pagoda's lower base is covered in reliefs of the twelve zodiac animals, and the upper base is decorated with portrayals of the eight legions. The first story of the pagoda's body displays carved depictions of the four heavenly guardian kings, displayed in sequence from the lowest rank at the bottom to the highest rank pictured in the upper-most position of the structure. Especially noteworthy are the zodiac animals, three portrayed on each side of the pagoda's lower base, as the subject is a rare sight on the bases of Korea's stone pagodas.[11] In comparison to

---

8   (Kim 2003b, pp. 89–109).
9   The five-story eastern pagoda of Hwaŏmsa also underwent restoration in October of 1999. However, as the western and eastern pagodas were most likely not conceived as twin pagodas, this paper focuses on the western structure as the sole research subject.
10  Though both the eastern and western pagodas were erected in the ninth century, based on the structures' styles and excavated relics, the construction of the latter preceded the former.
11  During the Unified Silla period, the twelve zodiac animals were typically portrayed in the stonework of royal tombs and are a representative feature of royal funerary art of the period. It is not uncommon for the same subject to appear on the base of pagodas as well, as it does on the Hwaŏm western pagoda. This cross appearance is considered the result of mutual influence between Buddhist sculpture and eighth-to-ninth-centuries' royal funerary sculpture. For more on this subject, see (Im 2018, pp. 153–78).

examples of the eighth century, the peak period of Unified Silla sculpture, the sculpture on the surface of the Hwaŏmsa western pagoda is rather unimpressive (Figure 6). The relative inadequacy of the elaboration and enshrinement of sarira in the first story of the pagoda's body are characteristic traits of ninth-century stone pagodas.[12]

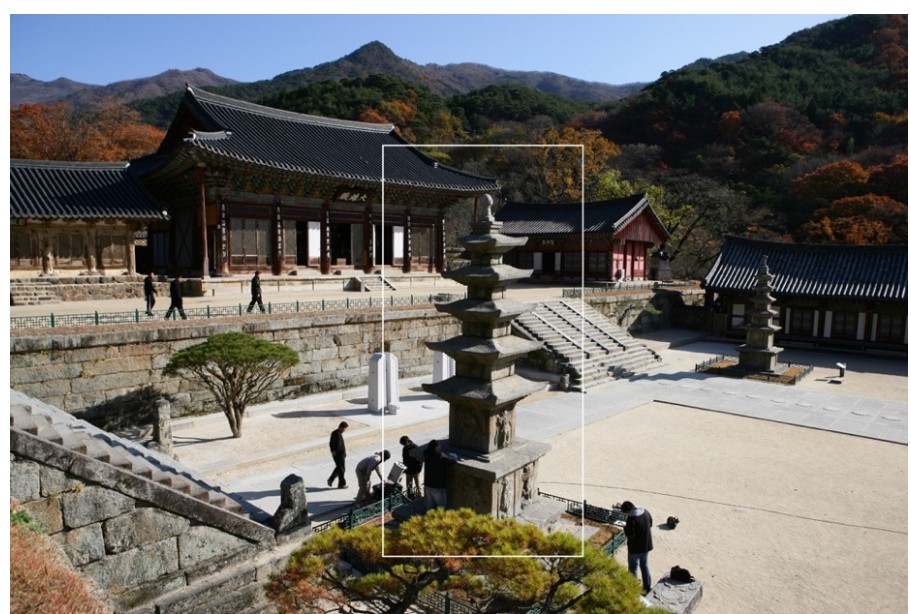

**Figure 4.** Two pagodas of Hwaŏm temple, Unified Silla, ninth century (**left**) western pagoda and (**right**) eastern pagoda [©Photograph by Lim, 2018].

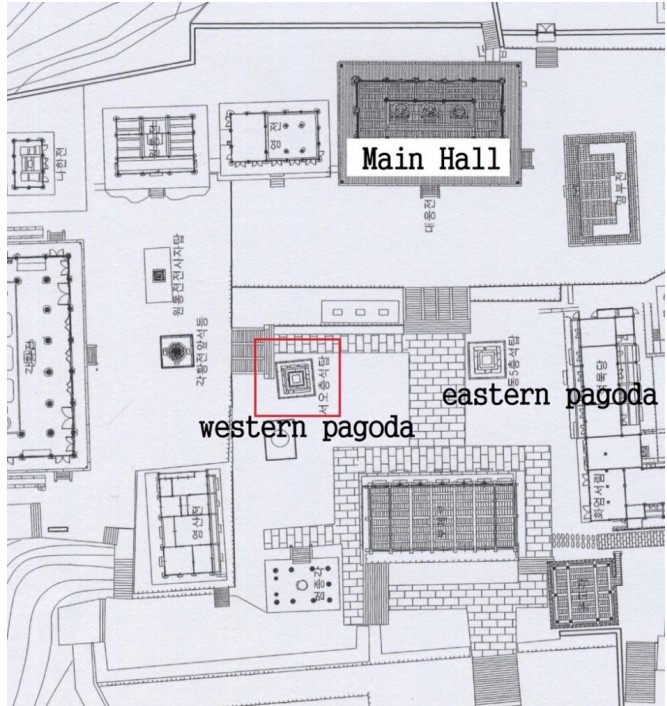

**Figure 5.** Floor plan of the two pagodas of Hwaŏm temple, square mark western stone pagoda [©Cultural Heritage Administration in Korea].

---

[12] In the case of ninth-century Unified Silla stone pagodas, the sarira was almost exclusively placed in the first story of the structure's body. (Pak 1994, pp. 96–97).

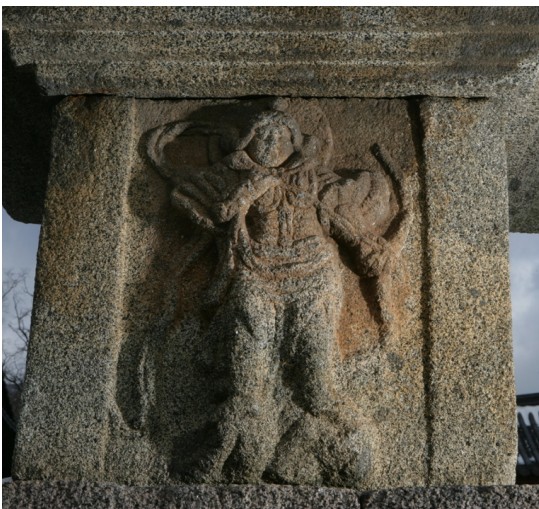

**Figure 6.** Four heavenly kings, Unified Silla, ninth century, first body of western stone pagoda, Hwaŏm temple [©Photograph by O Seyun, 2014].

Many relics were found in the ninth-century western pagoda of Hwaŏmsa.[13] The excavated relics were contained in the first story of the pagoda's body or the structure's base. Enshrined relics in the first story of the pagoda's body include a yellow glass bottle containing twenty-two of Buddha's sarira, a pair of celadon bottles, a bronze dish, two bronze lidded containers, *Wugoujingguang datuoluonijing* (無垢淨光大陀羅尼經, *The Great Dharani Sūtra*), and thirteen sheets filled with the stamped decoration of a pagoda, among other items (Figures 2, 7, and 8).[14] A bronze-lidded container and yellow glass bottle were encased in the celadon bottle that measured 16.5 centimeters in height. The bronze dish covered the celadon bottle's opening, an arrangement indicating that the celadon bottle functioned as the outer sarira casket—an unprecedented example of such a composition. *Wugoujingguang datuoluonijing* and pagoda-stamped sheets were found separately from but in proximity to the celadon bottle. The base of the pagoda contained the bronze Buddha mold that is the subject of this paper along with several other items, such as eleven bronze hairpins, two spoons, and various crystal balls.

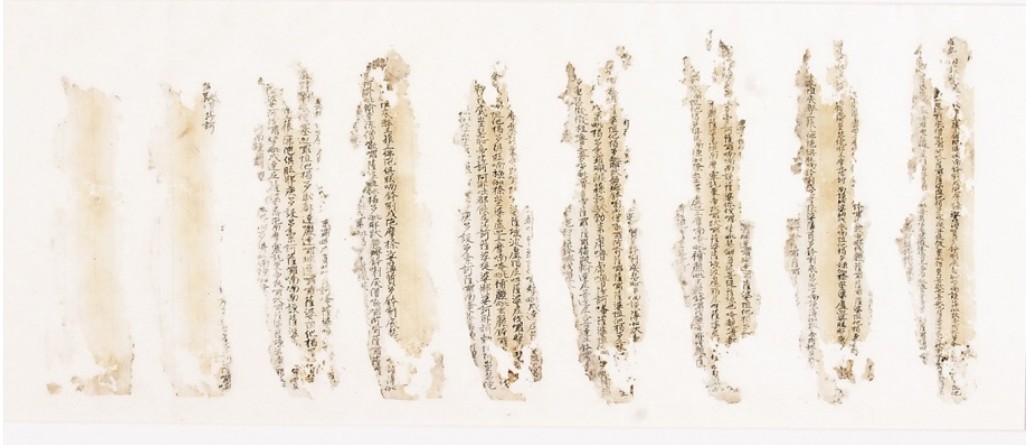

**Figure 7.** Pagoda-stamped sheets, Unified Silla, eighth-to-ninth centuries, H. 24.5 cm, Hwaŏmsa Buddhist Museum [©CBM].

---

13　(National Research Institute of Cultural Heritage 1997, pp. 162–67); For the most recent scholarship regarding the Buddha statue enshrined in the pagoda, see (I 2016, pp. 56–310; Kim 2016, pp. 39–71).

14　Unfortunately, the report for the 1995 excavation has yet to be published. For an introduction to the excavated relics, see (National Research Institute of Cultural Heritage 1997, pp. 162–67; Chu 2007, pp. 73–74; Ch'ae 2010, pp. 84–93).

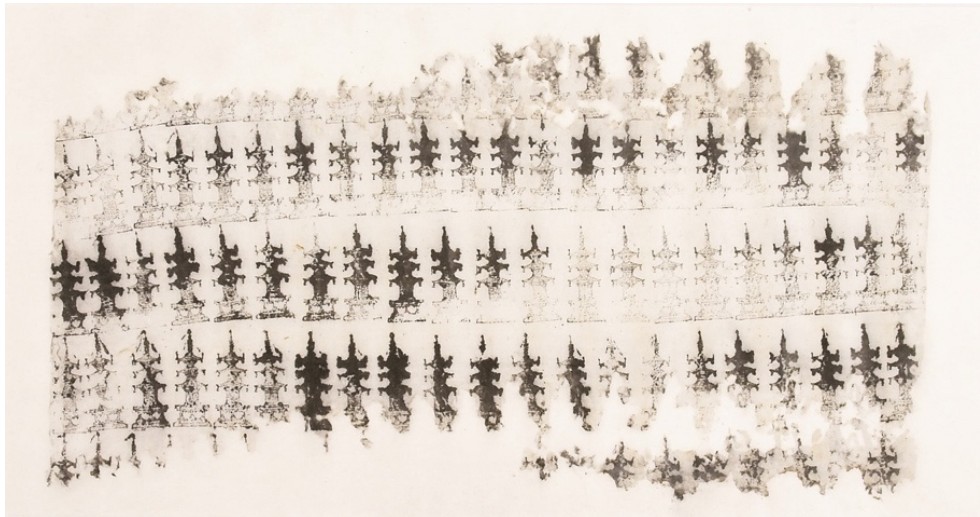

**Figure 8.** *Wugoujingguang datuoluonijing*, Unified Silla, eighth-to-ninth centuries, H. 24 cm, Hwaŏmsa Buddhist Museum [©CBM].

At this juncture, it is essential to examine the bronze Buddha mold's findspot by taking a closer look at *Wugoujingguang datuoluonijing* and the pagoda-stamped sheets that were found in the pagoda's body (Figures 7 and 8). *Wugoujingguang datuoluonijing* is a text concerned with the afterlife that was written by Mituo Shan 彌陀山 (fl. 704) and Facang 法藏 (643–712). The sūtra was written in 704 and made its way to the kingdom of Unified Silla in a mere two years. In 706, the sutra was enshrined in the three-story pagoda of the Hwangbok temple, constructed by royal decree in Kyŏngju.[15] Though it may not have been the first sūtra to arrive, *Wugoujingguang datuoluonijing* was introduced to the kingdom of Unified Silla relatively quickly and was largely popular during the ninth century.[16] The transcription of the sūtra enshrined in the Hwaŏmsa western pagoda is particularly distinct. This version is absent of the original text and contains only the dhāraṇī portion copied a total of twenty-eight times.[17] The estimated number of repetitions is based on the surviving fragments, but it is entirely possible that the complete version contained many more duplications. Moreover, the text was not printed from woodblocks; instead, it was handwritten by at least seven different individuals, judging from the discrepancies in handwriting (Figure 7).[18]

The principal doctrine of *Wugoujingguang datuoluonijing* asserts that "those who present various offerings according to the dhāraṇī way, during the process of constructing new or repairing old Buddhist pagodas, are granted longevity, rebirth in the pure land of the west, immunity from impairments and sins, and blessings in the afterlife."[19] The sūtra also describes the process of purging oneself of sin in order to receive blessings in the next life. These different methods are disclosed in detail through instructions such as "those who die at a young age or are riddled with many illnesses should repair old pagodas or construct a small pagoda from clay,"[20] and "set a square stone table in front of Buddha's pagoda and place various offerings on it."[21]

---

[15] The inscription on the lid of the sarira reliquary, dated to 706 and discovered in the three-story stone pagoda of the Hwangbok temple 皇福寺, states that it contains one transcription of *Wugoujingguang datuoluonijing*. However, contrary to the inscription, the text is missing and has most likely perished. The Hwangboksa reliquary is currently in the collection of the National Museum of Korea located in Seoul. For research on the Hwangboksa reliquary, see (Han 2006, pp. 61–88).

[16] (Chu 2004, p. 178).

[17] (Chŏng and Pucha 2016, p. 163).

[18] (Chŏng and Pucha 2016, p. 179).

[19] *Taishō* 19, no.1024:0718c20, 0720a07.

[20] *Taishō* 19, no.1024:0719b11, 0720c23.

[21] *Taishō* 19, no.1024:0718c01, 0720b09, 0720c23.

There is evidence that the rituals described in *Wugoujingguang datuoluonijing* were actually performed in Unified Silla during the ninth century. A sarira reliquary dated to 867 was excavated from the three-story western pagoda of Ch'uksŏ temple 鷲棲寺 in Ponghwa 奉化, Gyeongsangbuk-do (Figure 9). On the surface of the stone reliquary, the following phrase is inscribed: "a Buddhist ceremony from *Wugoujingguang datuoluonijing* was held [at the site of Ch'uksŏ temple in Ponghwa]. The monk in charge of the ceremony's teachings is Hyŏn'gŏ 賢炬 (fl. 867) of Hwangnyong temple 皇龍寺." From this excerpt, it is explicit that Buddhist ceremonies were executed through the creation of a square stone table and the performance of a ritual in the presence of a facilitator. It is particularly significant that the speaker who was requested and tasked with delivering the Buddhist teachings of the ceremony was Hyŏn'gŏ of Hwangnyong temple situated in the then capital city of Kyŏngju.[22] Despite the great distance of about 200 km separating Hwangnyong and Ch'uksŏsa in Ponghwa, the monk Hyŏn'gŏ was especially asked to oversee the Buddhist ceremony of *Wugoujingguang datuoluonijing* at Ch'uksŏsa. Thus, the reliquary text is evidence of the trips undertaken by Hwangnyongsa monks to the temples that maintained relationships with the royal court or to the large-scale temples in other provinces for the purposes of presiding over special events or delivering teachings in Buddhist ceremonies.

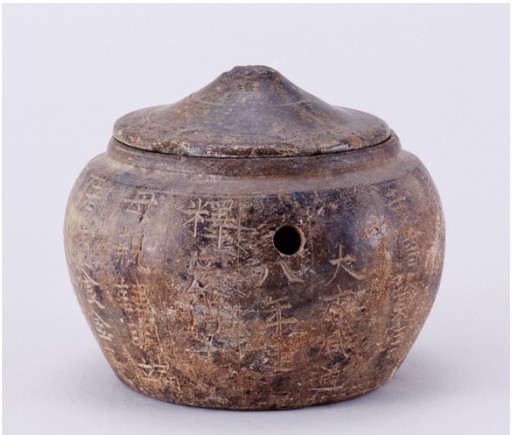

**Figure 9.** Sarira reliquary, Unified Silla, 867, 9.5 cm, Three-story western pagoda of Ch'uksŏ temple, in Ponghwa, Gyeongsangbuk-do, Korea, Kyŏngju National Museum, Korea [©Photograph by O Seyun, 2014].

It follows that Buddhist art produced as pious deeds according to *Wugoujingguang datuoluonijing* belongs to one of two categories. The first type is a square stone table placed in front of the pagoda, and the second type is in the form of 77 or 99 miniature pagodas enshrined inside the larger pagoda (Figure 10). An example of the former category is the rectangular stone situated in front of the ninth-century Hwaŏmsa 'Dharani pagoda' (the term 'Dharani pagoda' in this paper refers to any pagoda containing *Wugoujingguang datuoluonijing*). The rectangular stone was presumably placed and used as a square stone table for the ritual of *Wugoujingguang datuoluonijing* (note the stone object placed in front of the pagoda pictured in Figure 2). Forms resembling table legs are carved into the sides of the stone, which likely served as a table upon which various objects were placed. Nevertheless, though the stone object at the site of Hwaŏmsa most certainly functioned as a table, its role in the *Wugoujingguang datuoluonijing* Buddhist ceremony is but speculation.

---

[22] (Im 2017, pp. 41–75).

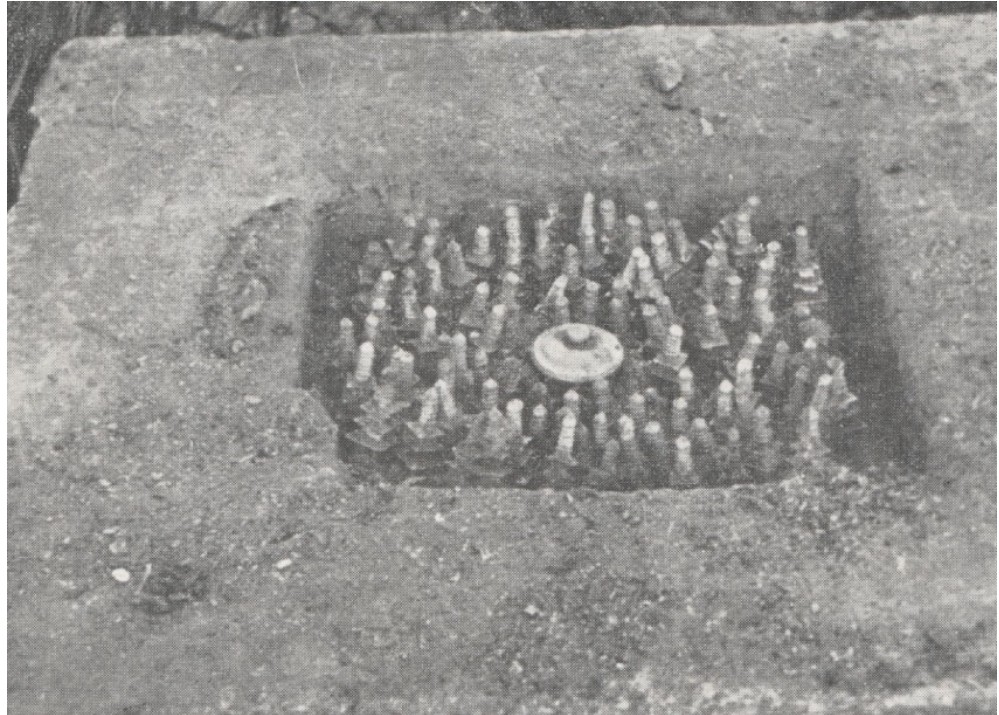

**Figure 10.** Miniature stupas in first story of three-story stone pagoda, Sŏdong-ri, in Ponghwa, Gyeongsangbuk-do, Korea (photograph 1962, at the time of excavation). [Source: (Hwang 1980, p. 6)], Unified Silla; miniature stupas are owned by Kyŏngju National Museum.

On the other hand, it is clear that the miniature pagodas discovered in the larger pagoda were meant for the ritual from *Wugoujingguang datuoluonijing*. Nearly all Dharani pagodas of the Unified Silla period contained miniature pagodas made of clay. The Hwaŏmsa western pagoda is an exception to this pattern. In place of the miniature clay pagoda, the Hwaŏmsa pagoda contained several sheets of paper covered with the stamped decoration of a small-scale version of the pagoda body in repetition (Figure 8).[23] Each pagoda motif is about five to six centimeters in height and was created by pressing an inked wooden stamp upon the paper's surface in a continuous fashion. The example found in the Hwaŏmsa western pagoda is the only one of its kind and is further interesting because of the inconsistent coloring of the repeated motifs, caused by the decreased amount of ink in each consecutive stamp. At the time of discovery, the stamped sheets were laid over the contents of *Wugoujingguang datuoluonijing* in a dry state,[24] undoubtedly signifying that the miniature pagoda motif is a representation of the object described in the sūtra itself.

The method of incising the design of a miniature pagoda into a piece of wood to create a stamped version of the motif most likely developed out of the desire to reduce the relatively time-consuming process of creating a three-dimensional model from clay. The resulting pagoda-stamped sheets were an integral component of the various items that were placed inside the pagoda along with *Wugoujingguang datuoluonijing*—objects enshrined only after the Unified Silla people of the eighth-to-ninth centuries had read and memorized the text countless times, copied it, and finally conducted the proper ritual. This process was carried out as a way of 'absolving one from sin,' 'accumulating meritorious deeds,' and ensuring blessings in the afterlife. *Wugoujingguang datuoluonijing* describes this process in comprehensive detail for the benefit of any individual who wished to replicate the proceedings.

---

[23] The miniature pagodas of *Wugoujingguang datuoluonijing* were primarily made of clay, but there are also examples made from stone, wood, and gilt bronze, among other materials. See (Han 2007, pp. 35–58).

[24] For scholarship on the preservation of the pagoda-stamped sheet and *Wugoujingguang datuoluonijing*, see (Pak 1997, pp. 83–111).

In commitment to the envisioned goal, the sūtra continuously emphasizes the act of 'repetition' or 'reproduction' as the most crucial aspect of the ritual.

The same is true of the bronze Buddha mold, as it is also an object created for the purpose of 'reproduction.' Held in one hand via the ring-shaped handle, the mold was used to 'stamp' the image of a Buddha onto the surface of prepared clay (Figure 1). The following section details an analysis of the mold itself. Additional objects to consider are the spoons and female hairpins that were excavated along with the bronze mold. Two spoons were discovered—one intact and the other partially fractured at some point (Figure 2). The undamaged spoon is 23.5 centimeters long, a length comparable to that of the spoons typically used in Korea today. The spoon's form indicates that it was for daily use during the mid-eighth century of the Unified Silla period, but the same form was used up until the mid-ninth century. Considering the spoon's form and the construction date of the Hwaŏmsa western pagoda, the spoon could have been produced in the early ninth century.[25] Excluding the ninth-century Yangsan 梁山 T'ongdo temple 通度寺 three-story stone pagoda, there is no other example of an ordinary spoon enshrined in a pagoda. It is unclear whether the spoon is an enshrined offering in and of itself or a utensil used in a Buddhist ritual and placed in the pagoda after the fact. The only verifiable fact is that spoons dating to the Unified Silla period have been predominantly discovered in sites related to the royal family.[26] In Unified Silla, the spoon was a precious object used by the upper class and thus a symbol of status. The same applies to the bronze hairpins used to hold a woman's hair in place (Figure 2). Eleven such hairpins were discovered inside the Hwaŏmsa pagoda. Though the bronze spoons and hairpins were both considered precious objects in Unified Silla, it should be noted that the lack of gold or silver versions of the objects renders them relatively humble and utilitarian in comparison to the other offerings enshrined in the pagoda.

Here, it is pertinent to question why the bronze Buddha mold, spoons, and hairpins were separated from the most important component—the sarira—and placed in the pagoda without protective casing. Some scholars regard the separately enshrined objects as items purposely left in the pagoda to be discarded, just as some Buddha statues are abandoned if judged to be below standard. From this perspective, these are not objects produced to be used in a sacred ritual that are coincidentally of inferior quality.[27] Whatever the case may be, the objects discovered in the pagoda's base are surely items acquired without a particular container. As mentioned previously, the pagoda itself was an accompaniment to the ritual of *Wugoujingguang datuoluonijing*, and, thus, the bronze mold, spoons, and hairpins enshrined at the base were assuredly offerings placed inside the structure as part of the ceremonial process.[28]

## 3. The Production Period and Characteristics of the Hwaŏmsa Bronze Buddha Mold

The Hwaŏmsa bronze Buddha mold measures 8.2 cm in length, 7.1 cm in width, and 0.4–0.5 cm in thickness. The circular ring perpendicular to the back of the mold is the handle. The mold was used as a stamp to make an impression in prepared clay, after which the resulting design was baked in a fire of increasingly high temperatures. Factoring in the inevitable shrinkage from high heat, the resulting clay Buddha would have been smaller than the mold—perfectly sized to be held in one hand. The image created by the mold is of a Buddha figure in Dharmacakra-mudrā that is seated on a lotus throne and flanked by a three-story pagoda on each side.

---

[25] (Chŏng 2008, pp. 312–13, 339). From Late Silla to Early Koryŏ, spoons were produced with increasingly smaller circular elements and significantly longer handles. An especially typical characteristic was the tapering width of the handle as it neared the circular element, which is in stark contrast to the shape of the Hwaŏmsa spoons.

[26] (Chŏng 2008, p. 359).

[27] (Chu 2007, pp. 79–80).

[28] For a view that is in agreement with this paper, see (Kim 2014, pp. 31–32).

The Chinese counterpart of these clay Buddhas started to become popular in the mid-seventh century.[29] As opposed to the term 'Jeonbul' 塼佛 (clay Buddha) used in Korea and Japan, the word 'Tuofo' 脱佛 meaning 'to remove from the mold' is the common name for these objects in China.[30] In *Datangxiyu qiufa gaosengchuan* 大唐西域求法高僧傳 (*An Account of Buddhism Sent from the South Seas*) by Yijing義淨 (635–713) of the Tang dynasty, there is an excerpt stating that "the East Indian king of the nation Sanmodazhaguo 三摩呾咤國 sends daily offerings of 10,000 clay Buddhas" in its records.[31] There are further mentions in another text written by Yijing in 691, *Nanhai jigui neifa zhuan*南海寄 歸內法傳 (*A Record of the Buddhist Religion as Practiced in India and the Malay Archipelago*), that states "in seventh-century India, offerings are made everywhere in the form of clay tablets made from molds or Buddha images stamped onto silk or paper. These objects are placed in a pile and surrounded by bricks to form stupas."[32] From these various records, it can be inferred that clay Buddhas were already being produced and used in mass quantities in the seventh century in places such as India.

The clay Buddhas that originated in India were very popular in China during the mid-seventh century after the return of the monk Xuanzang 玄奘 (602?–664). They were particularly high in demand in the city of Chang'an (presently Xi'an), the capital of China at the time.[33] Thus, the city of Xi'an is a rich discovery ground for clay Buddhas of the Tang dynasty; many have been found in such places as the Dacien temple 大慈恩寺 Dayanta 大雁塔, the Shiji temple 實際寺, the Liquan temple 醴泉寺, and the Ximing temple 西明寺. It is not a coincidence that these temples are widely associated with Xuanzang and also happen to be the most common discovery sites for Tang-dynasty clay Buddhas. There is a most probable connection between Xuanzang's return to China and the sudden popularity of the clay tablets. Other factors to consider are the low cost and simple method of manufacture, as both elements made the mass production of the clay Buddhas possible in the first place. Nevertheless, though Xuanzang played a major role in popularizing the clay Buddhas, the production of these objects in China actually predates the monk's return. The earliest known Chinese example of a clay Buddha comparable to the image of the Xingfu temple 興福寺 bronze Buddha mold is dated to the year 602 of the Sui dynasty (581–618). This Sui dynasty clay Buddha (Figure 11) produced in the Xingfu temple differs in style from the Hwaŏmsa mold but portrays the same basic image of a Buddha in Dharmacakra-mudrā flanked by twin pagodas. Many variations of this image, such as a Buddha in Dhyāna-mudrā 禪定印 or Bhūmisparśa-mudrā 降魔觸地印 surrounded by twin pagodas (Figures 12–14), exist in clay Buddhas of the seventh-to-eighth centuries. However, the Xingfu-temple clay Buddha of the Sui dynasty remains the earliest dated example to bear a close resemblance to the Hwaŏmsa mold.

---

[29] Though there are also parallel examples in regions such as India and Central Asia, this paper focuses on Chinese examples of direct relevance to the Hwaŏmsa bronze mold. The author wrote a separate paper titled *Extirpate Sins and Cultivate Goodness, Miniature Clay Buddha & Stupa* on the subject of molds and molded images from India and Central Asia that is currently in the process of publication. There is an extensive amount of academic research on the subject of Chinese clay Buddhas. The following texts are the most representative: (Goto 2008, pp. 1–71; Hida 2011, pp. 55–90; 2017, pp. 273–310).

[30] (Xiao 2015, pp. 94–125). In Tibet, the term 'tsha-tsha 擦擦' or 'cha-cha,' meaning 'reproduction,' is used to refer to clay Buddhas produced in the region. 'Tsha-tsha' is a transliteration of a word in the ancient Indian dialect of the north central region to the Tibetan language of Tibet, where the tradition of clay Buddhas was largely popular, as it still is today. The large-scale production of the clay Buddhas took place in Central Asia from 781 up until 848, a period during which Gansusheng 甘肅省 Hexi 河西 was under Tibetan control in Central Asia. (Li 2011; Zhang 2002, pp. 1–18).

[31] *Taishō* 51, no.2066:8; (Taddei 1970, pp. 70–86).

[32] *Taishō* 54, no.2125:226c.

[33] Though general consensus dictates that production of the clay Buddhas was local to the city of Chang'an in the Tang dynasty (Hajime 2002, p. 80; Goto 2008, p. 10), recent and ongoing discoveries of examples in cities such as Fufeng 扶風 have raised the idea that the objects' production was not limited to Chang'an and the surrounding area but occurred throughout all of East Asia.

Clay Buddhas portrayed holding their hands in Dharmacakra-mudrā were popular in East Asia from the early seventh century up until the latter half of the ninth century. Buddha images in this particular mudrā were produced in clay for about 200 years, including the image made by the Tang dynasty (618–907) monk Qiying 祁瑛 in 627, as well as the example with an inscription dated to 827 (Figures 12 and 14). The presence of twin pagodas on either side of a seated Buddha was also a prominent motif during this period. These elements persisted in the visual profile of the clay Buddhas for about 200 years because of the utilized production method. Because a large number of clay Buddhas were made with the same mold, there were few stylistic changes and one format persisted for an extensive amount of time. For this reason, the abundance of clay Buddhas found in Chang'an city temples that survived the Great Anti-Buddhist Persecution are, without exception, in the same image and format.

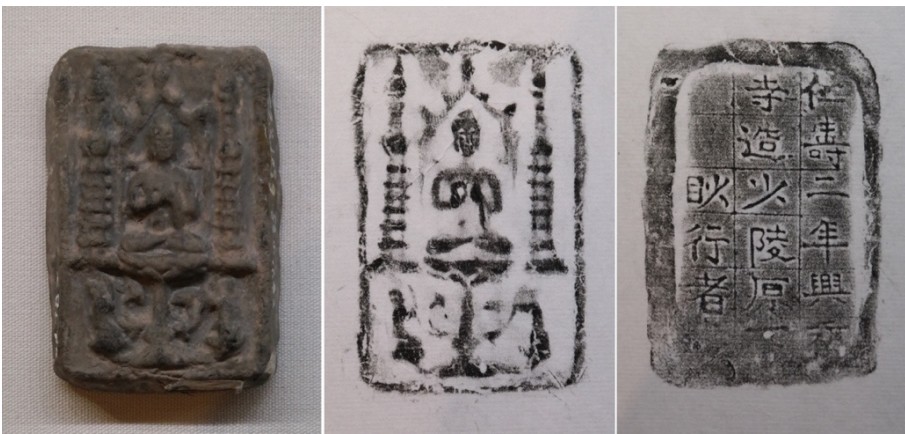

**Figure 11.** Clay Buddha (**left**), front rubbing (**middle**), and back rubbing (**right**). Sui Dynasty, 602, Xingfu temple, Xi'an, China, Beijing Capital Museum [©Photograph by Lim, 2017].

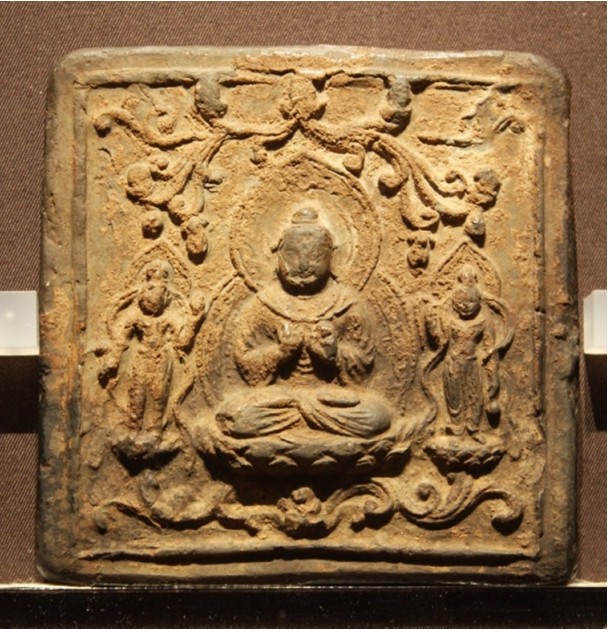

**Figure 12.** Clay Buddha, Tang Dynasty, ca. 627, 8.8 × 8.8 cm, Nara National Museum, Japan. [©Photograph by Lim, 2019].

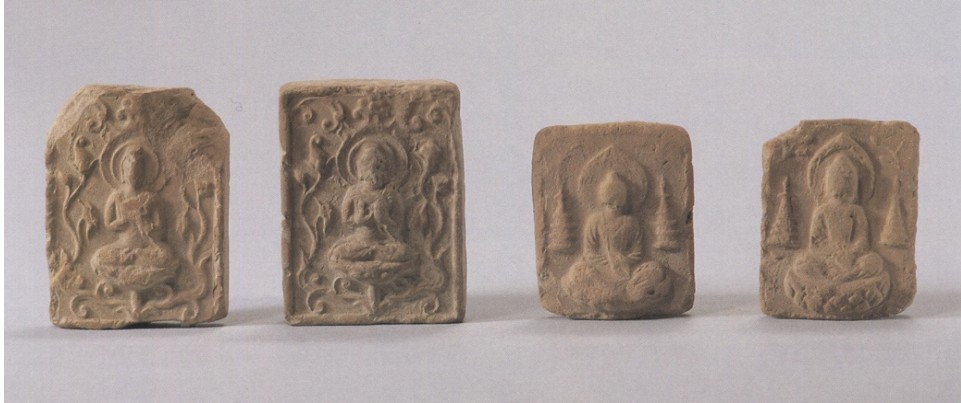

**Figure 13.** Clay Buddha, Tang Dynasty, Ximing temple, Xi'an, China, Zhongguo shehuikexueyuan kaoguyanjiusuo, [Source: (KKKFH 1995, plate 77)].

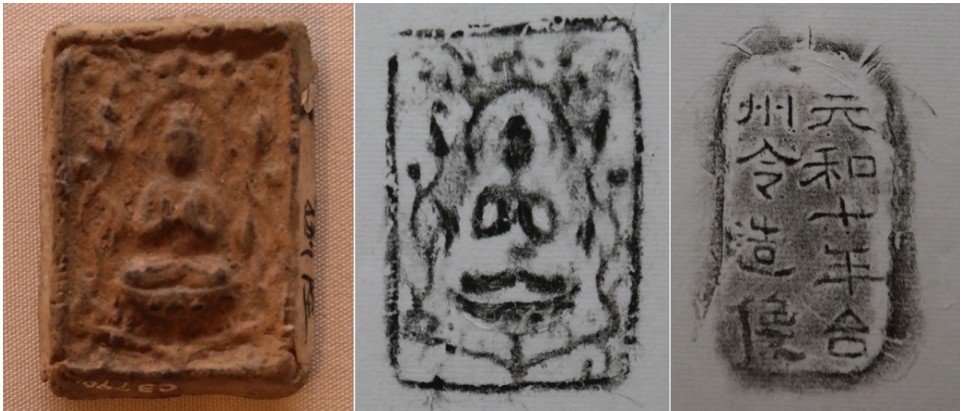

**Figure 14.** Clay Buddha (**left**), front rubbing (**middle**), and back rubbing (**right**). Tang Dynasty, 815, Beijing Capital Museum, China [©Photograph by Lim, 2017].

The image of a seated Buddha in Dharmacakra-mudrā was also a favored subject in Unified Silla and Japan during the latter half of the seventh century and into the eighth century. Some well-known examples are the two Wŏlchi 月池 (formerly Anapchi 雁鴨池) Buddha triad plaques dated to the late seventh century from Unified Silla (Figure 15), an Amitabha Buddha located on the sixth wall of the Horyu temple 法隆寺, and the five-figure relief of a seated Buddha from the Horyu Temple Treasures that is now in the Tokyo National Museum (Figure 16, image to the left).[34] The last Horyuji example is particularly pervasive, because the same five-figure image appears in large quantities and seemingly originates from one mold or several molds that were almost identical. This likeness is most evident in the large-scale five-figure Buddha relief excavated from the Niko temple 二光寺 in Nara by the Archaeological Institute of Kashihara Nara Prefecture in 2004–2005 (Figure 16, image to the right).[35] The Nikoji relief is especially important because it can be accurately dated to the year 694 from an inscription located at the bottom right portion, which reads 'Jiawu 甲午' year.[36] From the abundance of examples available to us, it is clear that clay Buddhas and reliefs identical to the image of clay Buddhas were prevalent in East Asia from the mid seventh century until the late ninth century. However the

---

[34] Wŏlchi was part of the royal palace located in Kyŏngju, the capital of Silla. The definite identification of the Japanese Buddha in Dharmacakra-mudrā as Amitābha is based on the presence of Avalokiteśvara and Mahāsthāmaprāpta at either of the Buddha's sides. Thus, it is highly possible that the Dharmacakra-mudrā signified a Buddha's identity as Amitābha in seventh-to-eighth-century East Asia.

[35] (Owaki 1986, pp. 4–25).

[36] (Nara National Museum 2015, p. 255).

molds themselves, such as the one from Hwaŏmsa, are extremely rare. Molds made in bronze rather than clay are still rarer, and aside from the Hwaŏmsa example, there are none to be found in all of East Asia.[37]

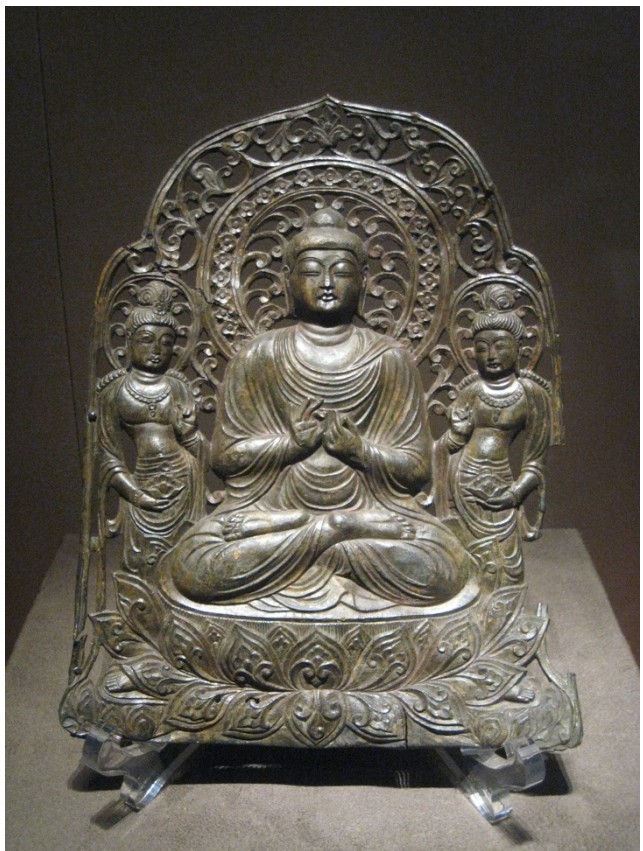

**Figure 15.** Gilt-bronze Buddha triad plaques. Unified Silla, late seventh century, H. 27 cm, National Treasure no.1475, Wŏlchi, Kyŏngju, Kyŏngju National Museum, Korea [©Photograph by Lim, 2019].

In general, there are two opposing views on the dating of the Hwaŏmsa bronze Buddha mold. Some scholars have dated the mold to the late seventh and early eighth centuries. Those who disagree with this theory have postulated the mold's production date to be in the late ninth-to-tenth centuries.[38] The two hypotheses are temporally separated by 200 years. The first theory indicating the late seventh and early eighth centuries focuses on the aforementioned Wŏlchi triad plaque of the Unified Silla period. The dating is based primarily on the consistency of the plaque's imagery, icon, and style. For the second theory, scholars have acknowledged that the Wŏlchi triad plaque is consistent in imagery and icon, but they have dated the mold to the late ninth and early tenth centuries, citing the lack of volume in style and secular quality of work. Of the two suggested dates, the former seems more probable. However, because of the longevity of a singular style corresponding to the method of production, it would be fitting to date the Hwaŏmsa bronze mold within a wider range encompassing the late seventh and eighth centuries.

---

[37]　There are no known examples of Buddha molds that date to the seventh and eighth centuries. However, there are a decent number of bronze Buddha molds concentrated in Southeast Asia dating to after the tenth century. (Goto 2008, pp. 98–117).

[38]　Scholars Kim Lena and Ch'oe Sŏna are advocates of dating the mold to the late seventh and early eighth centuries (Kim 2003a, pp. 228–32; 2007, pp. 68–70; Ch'oe 2004, p. 53), whereas Ch'oe Sŏngŭn dates the mold to the late ninth and early tenth centuries (Ch'oe 2000, p. 20).

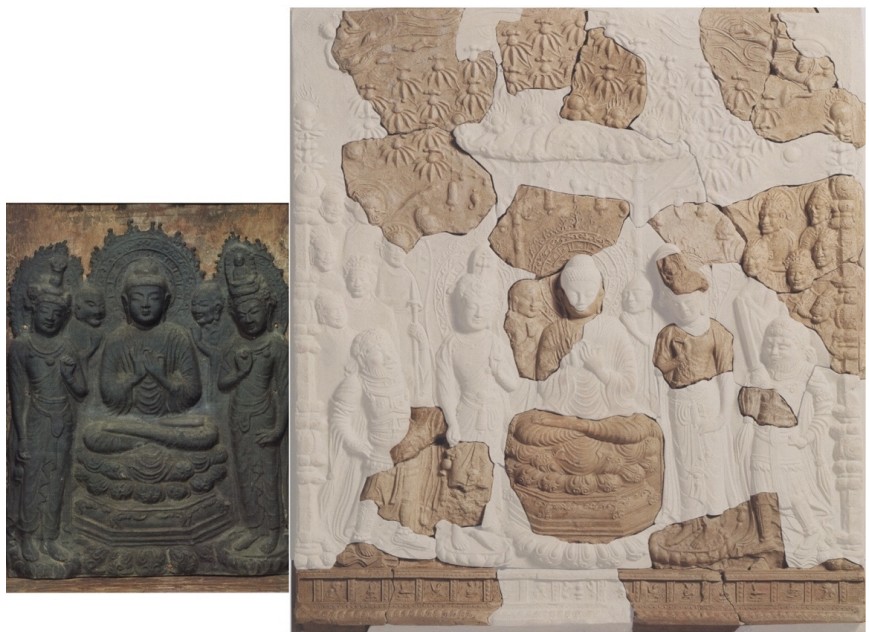

**Figure 16.** (**left**) Amitabha triad and two monks, 39.0 × 32.5 cm, seventh century; Tokyo National Museum. (**right**) Clay Buddha, Niko temple site, Nara, 50 × 50 cm, Kashihara kokogaku kennkyusho. [left Source: (NRDK 2001, plate 34), right Source: (Nara National Museum 2015, plate 93)].

The principal reason that there seems to be a disconnect between the Wŏlchi plaque and the Hwaŏmsa mold is the difference in size. The mold is 8 centimeters high, whereas the plaque is more than three times larger, at a height of 27 centimeters. Because the mold is significantly smaller, it inevitably lacks the details present in the plaque. Thus, the perceived differences in style actually result from their dissimilar dimensions rather than a disparate production period. Furthermore, the bulging uṣṇīṣa 肉髻 protruding from the Buddha's head is a representative characteristic of eighth-century Buddha images, and its presence indicates that production of the Hwaŏmsa mold cannot be dated to later than the eighth century. Further taking into account the eighth-century perception that a Buddha in Dharmacakra-mudrā signifies Amitabha, it is highly possible that the Buddha pictured in the mold is indeed Amitabha Buddha.[39]

An examination of historical context also points to the late seventh to eighth centuries as the most likely production period for the Hwaŏmsa mold. The inquiry begins with Wŏnch'ŭk 圓測 (613–696) of Silla. Wŏnch'ŭk was a monk who studied in China with Xuanzang and belonged to the order of the Ximing temple in Chang'an—a site where many clay Buddhas were discovered.[40] The Ximing temple was established in 656.[41] After it was completed in 658, Wŏnch'ŭk received the title of daedeuk 大德 (great virtuous) as well as an invitation to stay at the temple, after which he never returned to Silla. This was not by choice, because, although the thirty-first king of Unified Silla, Shinmunwang 神文王 (r. 681–692), requested Wŏnch'ŭk's return several times, Empress Wu Zetian 武則天 (r. 690–705) of China forbade Wŏnch'ŭk's departure. These circumstances are recorded in *Hwaŏmsajŏkki* 華嚴寺事跡記 (*History of the Hwaŏm Temple*), which was written in 1636 by Ch'oe Ch'iwŏn (857–?).[42] Ultimately, it was Tojŭng 道證 (620?–700?), a disciple of Wŏnch'ŭk, who returned alone to Unified Silla in 692 with a celestial map in his possession.[43] Tojŭng's arrival was a notable event for the people of Silla,

---

[39] Refer to footnote 33 of this text.
[40] (Cho 2010, pp. 357–88).
[41] *Datang daci'ensi sanzangfashi chuan* 大唐大慈恩寺三藏法師傳 *Taishō* 50, no.2053:0275b21; (ZSKXD 1990, pp. 45–55).
[42] (Ch'oe 1997).
[43] The Main History of Silla, Book 8, The first year, Kim bushik 金富軾, *Samguk sagi* 三國史記. King Hyoso 孝昭王 (r. 692–702) 元年.

who were acutely aware of Wŏnch'ŭk and his activities. Through Tojŭng, Wŏnch'ŭk and his beliefs or ideas strongly influenced Sillan society.[44] When Tojŭng returned to Silla with his celestial map, it is likely that he also had some clay Buddhas from the Chang'an temple that served as his residence in China. Not only were clay Buddhas largely popular in Chang'an, their portability made them easy to transport. Additionally, nearly all of the clay Buddhas discovered at the site of the Ximing temple in Chang'an portray Buddha in Dharmacakra-mudrā.[45] Of these, a majority also include the twin pagoda motif. Thus, it stands to reason that at least some, if not all, of the clay Buddhas that Tojŭng brought to Silla portrayed a Buddha figure in Dharmacakra-mudrā flanked by twin pagodas on either side. At the time, the clay Buddhas of the early Tang were undoubtedly a great influence on the Buddhist art of Unified Silla. The sum of all factors—Tojŭng's return to Silla after his studies in seventh-century Tang China, the period during which Buddha in Dharmacakra-mudrā was in vogue, the presence of a bulging uṣṇīṣa, among others—designates the eighth century as the most convincing production period for the Hwaŏmsa Buddha mold.

The production of clay Buddhas was ongoing in Unified Silla. Though aesthetically disparate from the Hwaŏmsa mold, the recently excavated Buddha image from the An'gang 安康 Kapsan temple site 甲山寺址 is about the same size, with a length of 8.3 centimeters, a width of 7.3 centimeters, and a thickness of 1.3 centimeters (Figure 17). The Kapsansaji Buddha is only one of many such examples made in Unified Silla. An informative analogy exists in the use of identical techniques to create both clay Buddhas and roof tiles during the Silla period. The copious number of roof tiles extant in Korea today further emphasizes the point that clay Buddhas were produced in mass quantities, especially considering the relative ease of using a mold. Nonetheless, it is important to distinguish the use of wooden molds in Old Silla from that of clay molds in Unified Silla during the production of roof tiles. Discoveries of clay molds in Japan dating to a similar period suggest that clay was also the preferred material for the Japanese Buddha mold (Figure 18). Considering both points, clay is the most plausible medium for the Buddha molds of Unified Silla. Thus, the Hwaŏmsa Buddha mold is a rarity, in that it is made of bronze. It is further unique because of its ring-shaped handle, which gives it the characteristic of being portable. The combination of the two irregularities allude to the possibility that the Hwaŏmsa mold was a special object that served another function in addition to the production of clay Buddhas.

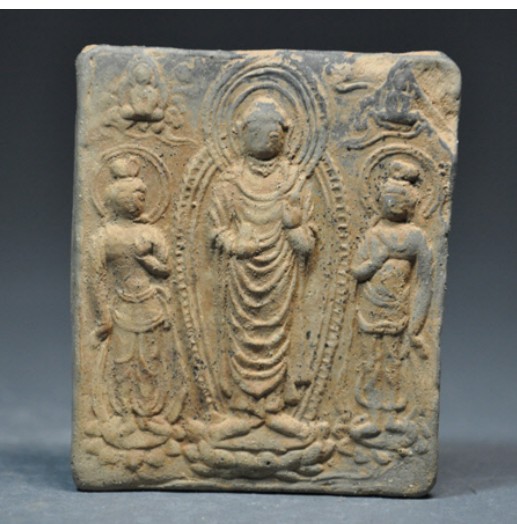

**Figure 17.** Clay Buddha, Unified Silla, 8.3 × 7.3 cm, An'gang Kapsan-ri temple site, Kyŏngju, Korea [©Photograph by Lim, 2016].

---

[44] Tojŭng, who preserved and advocated the story of Wŏnch'ŭk, passed on his beliefs and ideas to Taehyŏn—an important act that played a significant role in Silla Buddhism. (Chŏng 2007, p. 65).

[45] (ZSKXT 1990, pp. 45–55).

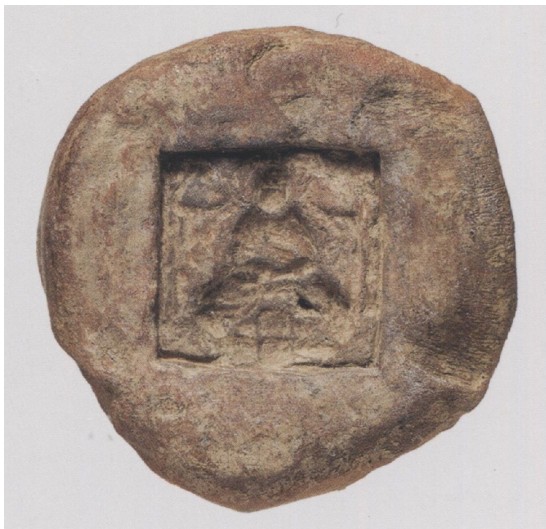

**Figure 18.** Clay mold, seventh century, 6.1 × 6.1 cm, Yamadadera temple site, Nara, Japan. [Source: (Nara National Museum 2015, plate 27-3)].

## 4. Status and Utility of the Hwaŏmsa Bronze Buddha Mold: A Ritual in Service of Dhāraṇī Offerings

The Chirisan Hwaŏm temple is cited as one of the top ten temples of Hwaŏm in *Pŏpchanghwasangjŏn* 法藏和尙傳 (*A Biography of Monk Facang written by Ch'oe ch'iwŏn* 崔致遠 (857–?) in 904.[46] Iryŏn 一然 (1206–1289) voices the same opinion in the thirteenth-century *Samguk Yusa* 三國遺事 (*Memorabilia of the Three Kingdoms*).[47] Both texts establish that Hwaŏmsa was one of the ten preeminent temples of the region. The publication year of Ch'oe ch'iwŏn's biography tells us that Hwaŏmsa maintained its high status during the ninth century, at the time of the western pagoda's construction.

In the ninth century, Hwaŏmsa was an official altar of precepts—a temple tasked with granting precepts to monks. It was a known kwandan sawŏn 官壇寺院, an official temple operated directly by the country for the purposes of carrying out *sukye* 受戒 or lay Buddhist ordinations. For example, the monk Sŏn'gaktaesa 先覺大師 Hyŏngmi 逈微 (864–917) studied at the Porim temple 寶林寺 but received his ordination in 882 at the Hwaŏm temple, as is recorded on Hyŏngmi's stele, despite the distance of 120 km between the two sites. Muwisa Sŏn'gaktaesa P'yŏn'gwangt'appi 無爲寺先覺大師遍光塔碑 (Memorial Stele for Supreme Master Sŏn'gaktaesa in Muwisa), designated Treasure number 507, is a monument that was erected in Hyŏngmi's honor in 946, twenty-nine years after his passing. Part of the inscription on the stele's surface reads: "When [Hyŏngmi] came to Hwaŏmsa in 882 to receive his precepts from the official altar, the Great Master took his seat upon the altar of precepts and at the same moment, energy of the white smoke completely filled the Buddhist sanctuary."[48] Another instance is the monk Kyŏngbo 慶甫 (869–947), who was ordained as a monk at the Puinsan temple 夫仁山寺[49] and practiced asceticism at the site of Paekkyesan 白鷄山 Tosūnghwasang 道乘和尙. In 886, Kyŏngbo traveled four kilometers from Paekkyesan to Hwaŏmsa to receive precepts, though he returned to his asceticism soon afterward.[50] It follows that in the ninth century, Hwaŏmsa was a central site of wide renown for its function as a temple that granted precepts to distinguished monks.[51]

---

[46]   *Taishō* 50, no.2054:0284c17. *Pŏpchanghwasangjŏn* is a record of the monk Facang's passing 192 years after its occurrence by Ch'oe ch'iwŏn.

[47]   *Taishō* 49, no.2039:1006c03.

[48]   Muwisasŏn'gaktaesap'yŏn'gwangt'appi 無爲寺先覺大師遍光塔碑 (http://gsm.nricp.go.kr/_third/user/viewer/viewer02.jsp).

[49]   Puinsan temple is known only by name and reputation; its location is yet unknown.

[50]   Ongnyongsadongjindaesabi 玉龍寺洞眞大師碑 (http://gsm.nricp.go.kr/_third/user/viewer/viewer02.jsp).

[51]   (Han 1988, p. 49; Pak et al. 2007, pp. 102–3).

Constructed in the ninth century, the western pagoda of Hwaŏmsa contained *Wugoujingguang datuoluonijing*, the pagoda-stamped sheets, and the Buddha mold. Though the pagoda also contained objects of the upper class, such as bronze spoons and hairpins, there was a notable lack of gold or silver objects among the relatively humble and utilitarian finds. The pagoda-stamped sheets and mold are also practical items that can be mass-produced. The creation of clay Buddhas cultivated high expectations due to the simplicity and low cost of production process as well as the short time required to manufacture large quantities. The act of creating a Buddha image, regardless of material, evokes blessings for the individual involved. *Fushuodacheng zaoxiang gongdejing* 佛説大乘造像功德經 (*Sūtra Spoken by the Buddha on the Merits of Image Making in the Mahāyāna*) written by Khotanese Tiyunbore 提雲般若 (fl. 689?–741?) in the late seventh century states the following on the subject of Buddha images as offerings:

"One uses materials such as red clay, white lime, and mud or wood—each to its strengths and within its limits—to create Buddha images. Though the image may be as small as a single finger, I will now illuminate you on the abundance of blessings the viewer can receive so long as the image is recognizable as Buddha."[52]

In other words, even if the image is made of clay rather than gold or silver, all individuals are eligible to receive blessings. The Hwaŏmsa mold is a special tool that generates Buddha images by the most economical means possible so that a wide range of users can rid themselves of sin and receive the protection of Buddha in both their current and subsequent lives. Concerning the function of the clay Buddhas, the following is an excerpt from a sūtra:

"In 692, there were two girls in Jiangzhou 降州 [of China]. They wished to achieve spiritual enlightenment by following a Buddhist nun's example of reciting the *Avataṃsaka Sūtra* 華嚴經. However, the nun suddenly passed away. For three years, the two girls visited the nun's burial place and paid their respects each morning by weeping aloud. After this period, an Indian monk appeared and, from her possessions, took out a clay image of about 18–20 centimeters. Bestowing the image upon the two girls, the monk said, "Present offerings to this image at your home. Then you will undoubtedly become monks in a short time." The girls did as advised and the image grew three centimeters every day until it reached a total height of 30 centimeters. ... Empress Wu Zetian heard about the wondrous occurrence, personally ordered the girls' heads to be shaved, and placed them in the Tiannu temple 天女寺."[53]

The above passage is important textual evidence communicating the narrative that the clay Buddhas were owned and used to fulfill the owner's wishes in late seventh-century China. A similar instance is the 'miezui 滅罪 (repentance)' seal in the hand of the eleven-headed Avalokiteśvara portrayed on the Tang dynasty Guangzhai temple 光宅寺 Qibaotai 七寶臺 of Chang'an that is currently in the collection of the Nara National Museum (Figure 19). The figure of the eleven-headed Avalokiteśvara stands at 110 centimeters and holds the seal in its right hand at shoulder height. The seal itself is a rectangular stamp that contains two Chinese characters, aligned vertically, that read 'miezui.' The eleven-headed Avalokiteśvara image is one side of the Qibaotai, or the Tower of Seven Treasures, commissioned by Empress Wu Zetian as a carving for the Guangzhai temple in Chang'an. It is a representation of the Tang people's desire to 'repent' and accumulate meritorious deeds in order to earn blessings.[54] Fulfilling this desire included the creation of seals used to repeatedly stamp a specific motif or phrase in order to join the world of Buddha. The acts of memorizing and transcribing dhāraṇī, as well as procuring inexpensive materials such as clay or paper to create stamped images of Buddha, were all effective methods that appealed to the masses of seventh and eighth century East Asia.

---

[52] *Taishō* 694, no.1:0793b24.
[53] *Dafangguangfu huayanjing ganyingchuan* 大方廣佛華嚴經感應傳. *Taishō* 51, no.2074.177b. Eds. Tang 唐 (618–907) Huiying 惠英 (?–?).
[54] *Shiyimian shenzhou xinjing* 十一面神呪心經. *Taishō* 20, no.1071:0152a21. Trans. Xuanzang 玄奘 (602?–664).

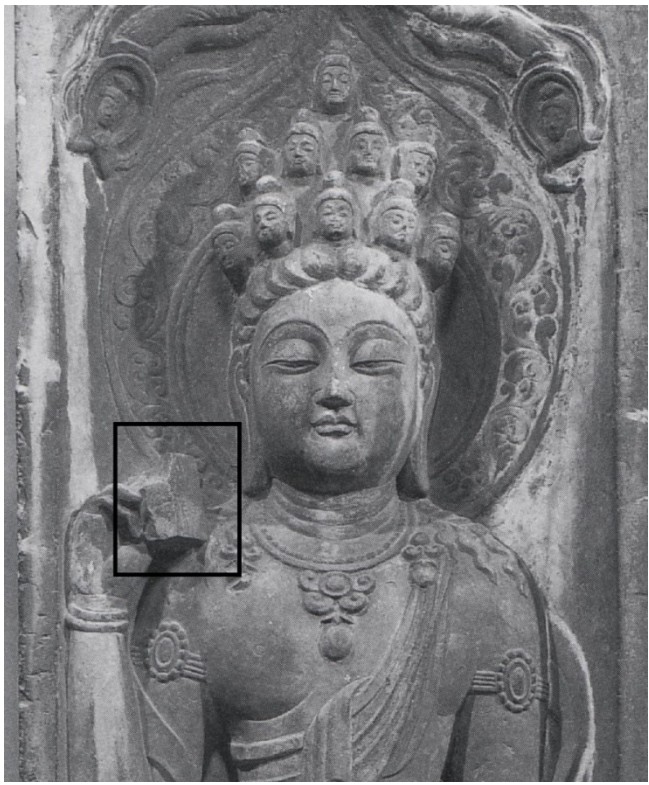

**Figure 19.** 'Miezui (repentance)' seal in the right hand of the eleven-headed Avalokiteśvara, Tang Dynasty, 703–704, Nara National Museum [Source: (Matsubara 1995, plate 655)].

Dhāraṇī was immensely favored in ninth-century Unified Silla and even used as a standard to select monks, who executed impressive recitations, as members of the taedŏk 大德 (Great Master) or senior monastics. According to Ch'oe Ch'iwŏn's records of the year 900, five of the twenty-eight taedŏk monks from Haein temple 海印寺were selected via recitation of dhāraṇī.[55] Thus, it seems that in the late Unified Silla period, monks that were experts on dhāraṇī were not restricted to a particular sect but present across a wide range of schools. Regardless of sect, dhāraṇī and, by extension, the dhāraṇī faith were integral criteria in the selection of the taedŏk and propagated throughout late Unified Silla.

With the acceptance of various dhāraṇī sūtras, the dhāraṇī faith spread gradually throughout Unified Silla from the late seventh century. Of the many sūtras, *Wugoujingguang datuoluonijing* was prominently favored because of its detailed but simple instructions. As a result, the construction of Dharani pagodas peaked in the ninth century due to the belief that following the guidelines outlined in *Wugoujingguang datuoluonijing* would absolve one of sin, bring blessings in the current life, and grant passage to the pure land of the west during the afterlife. Additional meritorious deeds included memorizing *Wugoujingguang datuoluonijing* and creating numerous miniature pagodas and Buddha images. The ideal subject for Buddha images was Amitābha.[56] Combined with the view that the Hwaŏmsa Buddha mold is a seventh and eighth century example of an East Asian image of Buddha in Dharmacakra-mudrā, a convincing argument can be made for the figure's identification as Amitābha.[57] At the Hwaŏm temple, the Buddha mold was the preferred way to gain the assistance of Amitābha to share among a large number of people, as it was mutually beneficial for the benefactor and the temple

---

[55] Ch'oe Ch'iwŏn 崔致遠. "Sillagayasanhaeinsaseonanjuwonbyeokgi" (新羅迦耶山海印寺善安住院壁記). *Dongmunseon* 東文選 64.

[56] For scholarship that examines the Amita faith of the late Unified Silla period through an analysis of *Wugoujingguang datuoluonijing*, see (Kim 1994, pp. 163–202).

[57] Refer to footnote 33 of this text.

leader. The mold's benefactor availed by performing a good deed at almost no expense in exchange for blessings, and the temple's leader gained believers and consequently monetary donations. If the benefactor did not wish to enshrine a clay Buddha in the temple, the image could also function as a sacred object in Buddhist services carried out at the benefactor's home.

Assistance from the region's inhabitants was essential to the construction of the western five-story pagoda of Hwaŏmsa. As an important ninth-century temple of the provinces rather than the capital area, Hwaŏmsa could not rely on the royal court as its sole financial supporter, and it was imperative for the temple to attract a large number of believers. The most straightforward way to accomplish such a task was for the temple monks to transcribe and memorize sūtras as well as to create miniature pagodas in tandem with the believers. For these purposes, the Chinese idea of employing clay Buddhas was a capable solution to the difficulties of producing Buddha images in large quantities—a challenge ninth-century Hwaŏmsa was able to overcome by using molds. Through the pursuit of 'repetition' encouraged by *Wugoujingguang datuoluonijing*, clay Buddhas provided the most accessible outlet for believers to unburden themselves of their sufferings. For this reason, the Hwaŏmsa bronze Buddha mold was a central implement in the ritual of *Wugoujingguang datuoluonijing* that was afterwards placed in the pagoda's base as an offering incumbent to the structure's construction.

## 5. Conclusions

Hwaŏmsa is one of many important large-scale temples that was built during the Unified Silla period. If Pulguksa and Hwangryongsa were representative examples of temples located in Kyŏngju, the capital city of the time, then the Hwaŏmsa and Haeinsa temples represented the surrounding provinces. As a kwandan sawŏn, Hwaŏmsa carried out lay Buddhist ordinations, a task that attributed elevated status to temples in the ninth century, the purported period of construction for the Hwaŏmsa western pagoda. Befitting a temple of high status, the relics enshrined in the western pagoda were accordingly unique and a contrast to those buried in other pagodas of the ninth century. In fact, the celadon sarira reliquary,[58] bronze Buddha mold, and pagoda-stamped sheets discovered in Hwaŏmsa's western pagoda are all one-of-a-kind in the whole of Korean Buddhist art. Moreover, the absence of gold or silver objects, which are otherwise frequent finds in royal temples, casts a humble light on the special albeit comparably modest relics of the Hwaŏmsa western pagoda.

The western pagoda is also a Dharani pagoda. This label signifies that benefactors of the pagoda placed *Wugoujingguang datuoluonijing* in the structure in the hopes of absolving their sins, accumulating good deeds, and receiving blessings in the afterlife. Of the various dhāraṇī, *Wugoujingguang datuoluonijing* is distinct for its detailed content and the simplicity of its instructions, which prescribes a transcription of the sūtra 77 or 99 times and creation of the same number of miniature pagodas. The anticipated results include freedom from suffering and entry to the pure land of the west. The Hwaŏmsa western pagoda does not contain a complete transcription of *Wugoujingguang datuoluonijing*; rather, it only contains the dhāraṇī portion, replicated 28 times. Furthermore, every transcription was produced not by a wooden block but directly by the hands of at least seven different people.[59] This act was not merely for the purposes of recitation but a deliberate process that was a part of a ritual to create a specific object to be enshrined in the Hwaŏmsa pagoda. The same circumstances apply to the pagoda-stamped sheets that were made by manually applying a wooden stamp to sheets of paper and the Hwaŏmsa bronze mold used to create clay images of a Buddha.

The eighth-century Hwaŏmsa bronze Buddha mold is of import to the study of religious art. Previous research on the function of religious art has demonstrated a concentration in examining images such as paintings or sculptures in addition to their influences on the masses. In contrast, this paper

---

[58] The act of using celadon as a sarira reliquary was not a rare occurrence during the Koryŏ 高麗 (918–1392) period. However, the celadon found in the pagoda of Hwaŏmsa is a singular example of the Unified Silla period.

[59] (Chŏng and Pucha 2016, p. 179).

looks at a specific example of a Korean religious art object in pursuit of a larger understanding in the universal nature of religion as reflected in art. Relevant Chinese examples of molds and molded images are adverted to as supplementary references in this study of the Hwaŏmsa mold. The sum of this paper is primarily a study of the Hwaŏmsa mold but simultaneously a preface to the further study of parallel examples found throughout Central and East Asia. By way of examining the Hwaŏmsa example, the function of Buddha molds in Korea and beyond become evident in their embodiment of the 'remembrance of Buddha,' personal ownership of a Buddha image, and collective ritual.

In a ritual, the facilitator would have pressed the Hwaŏmsa bronze Buddha mold into the clay held in the participants' hands to create a Buddha image meant to purge one of sins and grant blessings in the afterlife (Figure 20). Clay Buddhas are set apart from other Buddha images because they are made from a familiar material. Clay is inexpensive and thus conducive to producing clay Buddhas in large quantities. All individuals had equal opportunity to acquire ownership of a Buddha image at a reasonable cost. In addition, the length and width of the clay Buddhas rarely exceeded 10 centimeters, making them easy to transport and keep on one's person. The use of a mold rendered invalid any restrictions on the production of clay Buddhas. The creation of molded images was unrestricted by place or time and did not require any special techniques or skills. Furthermore, the Hwaŏmsa bronze Buddha mold was certainly produced in the eighth century for the purposes of ritual. Its function as a ritualistic object climaxed with the construction of the Dharani pagoda in the ninth century, which emphasized the concepts of repetition and replication based on *Wugoujingguang datuoluonijing*. Ultimately, the Hwaŏmsa bronze Buddha mold was a means for both the benefactors and users to draw closer to the Buddha.

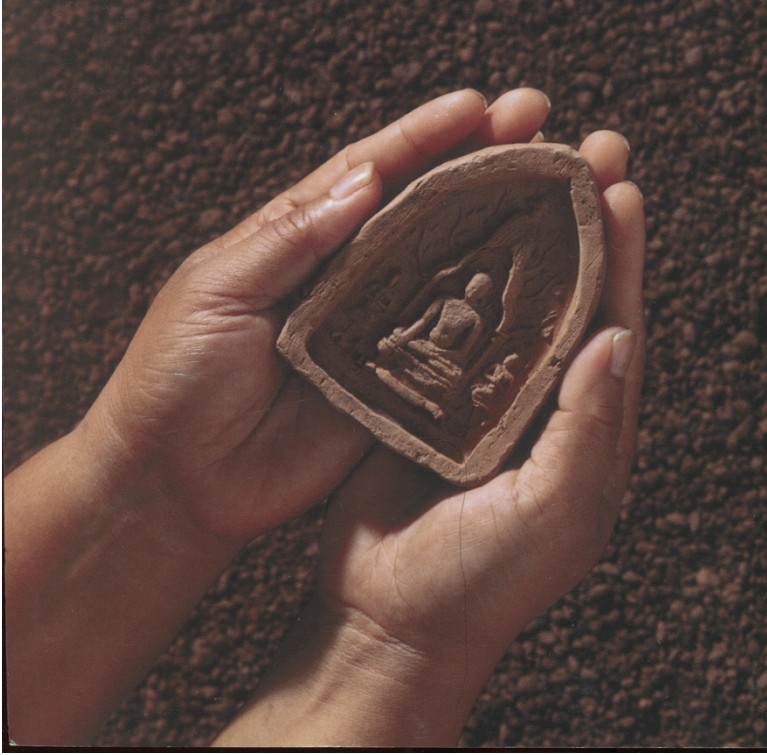

**Figure 20.** Clay Buddha, Pagan period, 12th–13th centuries, 9.6 × 7.1 cm, Myanmar [Source: (Fukuoka Art Museum 2008, cover picture)].

**Funding:** The research received funding from the National Research Foundation of Korea.

**Acknowledgments:** This work was supported by the Ministry of Education of the Republic of Korea and the National Research Foundation of Korea (NRF-2019S1A5A2A01045345).

**Conflicts of Interest:** The author declares no conflict of interest.

## Abbreviations

CBM Han'gukpulgyojungangbangmulgwan (韓國佛敎中央博物館) Central Buddhist Museum in Korea
GNRICH Gyeongju National Research Institute of Cultural Heritage.
KKKFH Kashihara kokogaku kennkyu fuzoku hakubutsukan (橿原考古學研究所附屬博物館). 1995. *Kentoshi ga mita chu kuni bunka* 遣唐使が見た中國文化. Nara.
NRDK Nara rokudaitera daimi kankokai 奈良六大寺大觀刊行會. 2001. *Nara rokudaitera daimi* 奈良六大寺大觀 volume 3, Hōryū-ji 法隆寺 3. Tokyo: Iwanami Shoten Publishers.
T Takakusu Junjirō 高楠順次郎 and Watanabe Kaikyoku 渡辺海旭 et al., eds., 1924–1935. *Taishō shinshū dai zōkyō* 大正新脩大藏經. Tokyo: Taishō issaikyō kankōkai 大正一切経刊行会.
ZSKXT Zhongguo Shehuikexueyuan Kaoguyanjiusuo Xi'an Tangchengdui (中國社會科學院考古研究所西安唐城隊). 1990. Tang chang'an ximingsi yizhi fajue jianbao (唐長安西明寺遺址發掘簡報). *Kaogu* (考古) 1990–1991.

## References

### *Primary Sources*

*Dafangguangfu huayanjing ganyingchuan* 大方廣佛華嚴經感應傳 *Taishō* 51, no.2074.
*Datang daci'ensi sanzangfashi chuan* 大唐大慈恩寺三藏法師傳 *Taishō* 50, no.2053.
*Datangxiyu qiufa gaosengchuan* 大唐西域求法高僧傳 *Taishō* 51, no.2066.
*Dongmunseon* 東文選.
*Fushuodacheng zaoxiang gongdejing* 佛説大乘造像功德經 *Taishō* 694, no.1.
*Hwaŏmsajŏkki* 華嚴寺事跡記.
Muwisasŏn'gaktaesap'yŏn'gwangt'appi 無爲寺先覺大師遍光塔碑.
*Nanhai jigui neifa zhuan* 南海寄歸内法傳 *Taishō* 54, no.2125.
Ongnyongsadongjindaesabi 玉龍寺洞大師碑.
*Pŏpchanghwasangjŏn* 法藏和尚傳 *Taishō* 50, no.2054.
*Samguk Yusa* (三國遺事 Legends and History of the Three Kingdoms) *Taishō* 49, no.2039.
*Shiyimian shenzhou xinjing* 十一面神呪心經 *Taishō* 20, no.1071.
*Wugoujingguang datuoluonijing* 無垢淨光大陀羅尼經 *Taishō* 19, no.1024.

### *Secondary Sources*

Ch'ae, Haechŏng (蔡勁廷). 2010. *Hwaŏmsaŭi Pulgyogongye* (華嚴寺의 佛敎工藝). *Hwaŏmsaŭi Pulgyomisul* (華嚴寺의 佛敎美術). Pulgyomisuryŏn'gu chosabogo (佛敎美術研究 調查報告) 2. Seoul: National Museum of Korea 國立中央博物館, pp. 84–93.
Ch'oe, Ch'iwŏn 崔致遠. 1997. Hwiilmun 諱日文. In *Hwaŏmsaji* 華嚴寺誌. Seoul: Aseamunhwasa, pp. 62–63.
Ch'oe, Sŏna (崔善娥). 2004. Tongashia 7–8segi chŏnbŏmnyunin amit'abulchwasang yŏn'gu (東아시아 7–8世紀 轉法輪印 阿彌陀佛坐像 研究). *Misulsahakyŏn'gu* (美術史學研究) 244: 53.
Ch'oe, Sŏngŭn (崔聖銀). 2000. Hwaŏmsa sŏoch'ŭngsŏkt'apch'ult'o ch'ŏngdongje pulsangt'ŭre taehan koch'al (華嚴寺 西五層石塔出土 靑銅製 佛像틀(范)에 對한 考察). *Kangjwa misulsa* (講座 美術史) 15: 24–46.
Cho, Kyŏngch'ŏl (조경철). 2010. Shilla wŏnch'ŭgŭi saengaee taehan kŏmt'o -shinbun'gwa kwigungmunjerŭl chungshimŭro (新羅 圓測의 生涯에 對한 檢討 –身分과 歸國問題를 中心으로). *Han'gukkodaesayŏn'gu* (韓國古代史研究) 57: 357–88.
Chŏng, Kyŏngchae (정경재), and Pak Pucha (朴富子). 2016. Hwaŏmsa sŏoch'ŭngsŏkt'am palgyŏn *Mugujŏnggwangdaedarani* ŭi sŏjijŏk yŏn'gu (華嚴寺 西五層石塔 發見『無垢淨光陀羅尼』의 書誌的 研究). *Sŏjihakyŏn'gu* (書誌學研究) 65: 163.
Chŏng, Pyŏngsam (鄭炳三). 2007. Shilla kubŏpsŭngŭi kubŏpkwa chŏndo (新羅 求法僧의 求法과 傳道). *Pulgyoyŏn'gu* (佛敎研究) 27: 65.
Chŏng, Ŭito (鄭義道). 2008. Ch'ŏngdongsutkaragŭi tŭngjanggwa hwaksan samgukshidae~t'ongilshillashidae (靑銅숟가락의 登場과 擴散: 三國時代~統一新羅時代). *Sŏktangnonch'ong* (石堂論叢) 42: 312–13, 339.
Chu, Kyŏngmi (周炅美). 2004. Han'gung pulsarijangŏme issŏsŏ *Mugujŏnggwangdaedaranigyŏn*ŭi ŭiŭi (韓國 佛舍利 莊嚴에 있어서『無垢淨光大陀羅尼經』의 意義). *Pulgyomisulsahak* (佛敎美術史學) 2: 178.

Chu, Kyŏngmi (周炅美). 2007. *Hwaŏmsasŏoch'ŭngsŏkt'apch'ult'o Sarijangŏmguŭi Koch'al* (華嚴寺西五層石塔出土舍利莊嚴具의 考察). *2007 Sangsŏlchŏn (2007* 常設展*)*. Seoul: Pulgyojungangbangmulgwan 佛教中央博物館, pp. 73–74.

Fukuoka Art Museum. 2008. *Buddha in the Palm of Your Hand—Clay Votive Tablets of Mainland Southeast Asia (*掌のほとけ—インドシナ半島の塼佛*)*. Fukuoka: Fukuoka Art Museum, 圖22, 27. pp. 98–117.

Goto, Munetoshi (後藤宗俊). 2008. *Senbutsu No Kita Michi-Hakuho-ki Bukkyo Juyo No Yoso (*塼佛の來た道-白鳳期佛教受容の樣相-*)*. Kyoto: Shibunkakushuppan (思文閣出版), pp. 1–71.

Hajime, Hagiwara (萩原 哉). 2002. Genzo hotsugan jukotizo ko -To zengyodoro senbutsu o megutte- (玄奘發願'十俱胝像'考—'唐善業泥'塼佛をめぐって). *Bukkyo gei-jutsu (*佛教藝術*)* 261: 80.

Han, Chŏngho (韓政鎬). 2006. Kyŏngju kuhwangdong samch'ŭngsŏkt'am sarijangŏmguŭi chaejomyŏng (慶州 九黃洞 三層石塔 舍利莊嚴具의 再照明). *Misulsanondan (*美術史論壇*)* 22: 61–88.

Han, Chŏngho (韓政鎬). 2007. Shilla mugujŏngsot'am yŏn'gu (新羅 無垢淨小塔 研究). *Tongangmisulsahak (*東岳美術史學*)* 8: 35–58.

Han, Kimun (한기문). 1988. Shillamal·koryŏch'oŭi kyedansawŏn'gwa kŭ kinŭng (新羅末●고려초의 계단사원과 그 기능). *Yŏksagyoyungnonjip (*歷史教育論集*)* 12: 49.

Hida, Romi (肥田 路美). 2017. Sŏan ch'ult'o chŏnburŭi chejang paegyŏnggwa ŭiŭi (西安 出土 塼佛의 製作 背景과 意義). *Kangjwamisulsa (*講座美術史*)* 48: 273–310.

Hida, Romi (肥田路美). 2011. *Hatsuto Bukkyo Bijutsu No Kenkyu (*初唐佛教美術の 研究*)*. Tokyo: Chuokoron bijutsu shuppan (中央公論美術出版), pp. 55–90.

Hwang, Suyŏng (黃壽永). 1980. Chŏnbung kimjech'ult'o paekchedongp'anbulsang (全北 金堤出土 百濟銅板佛像). *Pulgyomisul (*佛教美術*)* 5: 3–9.

I, Punhŭi (李芬熙). 2016. Han'gung sŏkt'am pulsang pongan yŏn'gu (韓國 石塔 佛像 奉安 研究). Ph.D. dissertation, Tongguk University, Seoul, Korea; pp. 56–310.

Im, Yŏngae (Lim, Young-ae). 2018. The 'Lion and Kunlun Slaves' Image: A Motif of Buddhist Art Found in Silla Funerary Sculpture. *Sungkyun Journal of East Asian Studies* 18–22: 153–78.

Im, Yŏngae (Lim, Young-ae 林玲愛). 2017. Ponghwa ch'uksŏsa sŏkchobirojanabulchwasang min mokchogwangbae (奉化 鷲棲寺 石造毘盧遮那佛坐像 및 木造光背). *Ihwasahakyŏn'gu (*梨花史學研究*)* 55: 41–75.

KKKFH—Kashihara kokogaku kennkyu fuzoku hakubutsukan (橿原考古學研究所附屬博物館). 1995. *Kentoshi ga mita chu kuni bunka (*遣唐使が見た中國文化*)*. Nara: KKKFH, plate 77.

Kim, Ch'uyŏn (金聖蝴). 2016. T'ongilshilla~koryŏ chŏn'gi t'amnae pongan pulsangŭi sŏngnipkwa paegyŏng (統一新羅~高麗 前期 塔內 奉安 佛像의 成立과 背景). *Misulsayŏn'gu (*美術史研究*)* 30: 40–69.

Kim, Lena. 2007. *Buddhist Sculpture of Korea*. Seoul: Hollym, pp. 68–70.

Kim, Rina (Kim, Lena 金理那). 2003a. *Han'gung Kodae Pulgyojogang Pigyo Yŏn'gu (*韓國 古代 佛教彫刻 比較 研究*)*. Seoul: Munyech'ulp'ansa, pp. 228–32.

Kim, Sanghyŏn (金相鉉). 2003b. Hwaŏmsaŭi ch'anggŏn shigiwa kŭ paegyŏng (華嚴寺의 創建時期와 그 背景). *Tongguksahak (*東國史學*)* 37: 89–109.

Kim, Tongha (金東河). 2014. Shilla t'ongilgi sogŭmdongbulsangŭi yuhyŏnggwa yongdoe kwanhan shiron (新羅 統一期 小金銅佛像의 類型과 用途에 關한 試論). *Pulgyomisulsahak (*佛教美術史學*)* 18: 9–10.

Kim, Yŏngmi (金英美). 1994. *Shillahadaeŭi Amit'ashinang (*新羅下代의 阿彌陀信仰*)*. *Shilla Bulgyo Sasangsa Yŏn'gu* (新羅佛教思想史研究). Seoul: Minjoksa, pp. 163–202.

Li, Ling (李翎). 2011. Caca yu shanyeni kaobian (擦擦与善业泥考辨). *Zhongguo guojia bowuguan guankan (*中国国家博物馆馆刊*)* 6: 110–28.

Matsubara, Saburo. 1995. *Chukuni Bukkyo Chokokushiron (*中國佛教彫刻史論*) 3*. Tokyo: Yoshikawakobunkan, plate 655.

Nara National Museum (奈良國立博物館). 2015. *Hakuho-Hanahiraku Bukkyo Bijutsu- (*白鳳-花ひらく佛教美術-*)*. Nara: Nara National Museum, p. 255.

NRDK—Nara rokudaitera daimi kankokai (奈良六大寺大觀刊行會). 2001. *Nara rokudaitera daimi* 奈良六大寺大觀. Hōryū-ji 法隆寺 3. Tokyo: Iwanami Shoten Publishers, Volume 3.

National Research Institute of Cultural Heritage (國立文化財研究所). 1997. Kurye hwaŏmsa sŏoch'ŭngsŏkt'ap(pomul che 133ho) ch'ult'o yumurŭi pojonch'ŏri (求禮 華嚴寺 西五層石塔(寶物 第133號) 出土 遺物의 保存處理). *Pojon'gwahakyŏn'gu (*保存科學研究*)* 18: 162–67.

Owaki, Kiyoshi (大脇 潔). 1986. Senbutsu to oshidashi hotoke no do genkei shiryo -Natsumi hiji no senbutsu o chushin to shite (塼佛と押出佛の同原型資料-夏見廢寺の塼佛を中心として). *MUSEUM* 418: 4–25.

Pae, Chintal (裴珍達). 2003. *Tangdae Pulgyojogak (*唐代 佛敎彫刻*).* Seoul: Ilchisa, pp. 220–29.

Pak, Chisŏn (朴智善). 1997. Hwaŏmsa sŏoch'ŭngsŏkt'am ch'ult'o chiryuyumul pojonch'ŏri (華嚴寺 西五層石塔 出土 紙類遺物 保存處理). *Pojon'gwahakyŏn'gu (*保存科學研究*)* 18: 83–111.

Pak, Kyŏngsik (朴慶植). 1994. *T'ongilshilla Sŏkchomisul Yŏn'gu (*統一新羅 石造美術 研究*).* Seoul: Hakyŏnmunhwasa, pp. 96–97.

Pak, Ŏnkon (박언곤), Chaein (이재인) I, and Hyosik (최효식) Ch'oe. 2007. Han'gung pulgyosawŏnŭi kyedan'gwa kyedandogyŏngŭi pigyo yŏn'gu (韓國 佛敎寺院의 戒壇과 戒壇圖經의 比較 研究). *Kŏnch'ukyŏksayŏn'gu* 建築歷史研究 16–22: 102–3.

Taddei, Maurizio. 1970. Inscribed Clay Tablets and Miniature Stūpas from Gaznī. *East and West* 20: 70–86.

Xiao, Guiian (肖貴田). 2015. *Paektobulgwa T'albure Taehan Koch'al* (白陶佛과 脱佛에 對한 考察). *Kukchehaksulshimp'ojiŏm Charyojip, Kodae Pulgyojogagŭi Hŭrŭm* (國際學術 심포지엄 資料集, 古代 佛敎彫刻의 흐름). Seoul: National Museum of Korea, pp. 94–125.

Zhang, Jianlin (張建林). 2002. Zangchuan fojiao caca gailun (藏傳佛敎擦擦概論). In *Zhongguo Zangchuan Fojiao Diaosu Quanji (*中國藏傳佛敎雕塑全集*) 4: Chacha* 擦擦. Edited by Zhongguo Zangchuan Fojiao Diaosu Quanji Bianji Weiyuanhui (中國藏傳佛敎雕塑全集編輯委員會). Beijing: Beijing Meishu Sheying Chubanshe, pp. 1–18.

ZSKXT—Zhongguo Shehuikexueyuan Kaoguyanjiusuo Xi'an Tangchengdui (中國社會科學院考古研究所西安唐城隊). 1990. Tang Chang'an Ximingsi Yizhi Fajue Jianbao (唐長安西明寺遺址發掘簡報). *Kaogu (*考古*)* 1990–1991: 45–55.

