# Peer review of "Buddhist Ritual and the Bronze Buddha Mold Excavated from the Western Five-Story Stone Pagoda of Hwaŏm Temple, Korea"

_religions, doi:10.3390/rel11030141_

Round 1

Reviewer 1 Report

Review of Buddhist Ritual and the Bronze Buddha Mold:

• Interesting topic which is of import to Buddhist studies beyond Korea. However, the author seems to be unusually fixated on Korea (with the exception of the final section). This could be remedied by some restructure of the text.

• The language needs to be effectively ‘beefed-up,’ it is rudimentary and somewhat inadequate.

• There is a tendency that the author takes knowledge of the various Korean sites referred to in the article for granted. This should be corrected so that adequate information is being provided for all sites, i.e. ref. to relevant articles or books. Korean Buddhism and its history are not widely known outside the field.

• Since Hwaŏm Temple is central to the article, I would suggest that some background to this important temple be provided, esp. for the Silla period.

• The article uses the silly Korean manner in referring to Buddhist temples by applying a double mode, i.e. ‘Hwaŏmsa Temple’, which then reads: ‘Hwaŏmsa Temple Temple.’ Silly, no?

• In general I would discuss the use of molds and molded images within the larger context of Central and East Asian Buddhism. The practice is known from India to China and on to Korea and Japan. There are interesting, early examples from the Silk Road. At the end of the article the author extends her or his discussion to mainly Chinese examples of the Tang. That is a good thing, but it could in principle be extended even further.

• Formally Sanskrit names should be used for Sanskrit works, and Chinese for Chinese works. Mixing them up or providing titles in Korean transcription should not be accepted for non-Korean works.

• There are a number of transcription mistakes, fx. the author mixes up two different Chinese transcription systems, eg. I-tsing when it should be Yijing. They should be corrected.

Conclusion:

Interesting paper with a lot of interesting data, but it requires a thorough, editorial brush-up to merit publication.

Author Response

RESPONSE TO REVIEWER 1

Point 1: Interesting topic which is of import to Buddhist studies beyond Korea. However, the author seems to be unusually fixated on Korea (with the exception of the final section). This could be remedied by some restructure of the text.
Response 1: This paper studies a universal characteristic of religion through the primary and specific examination of Hwaŏmsa in Korea. Naturally, Korea is the focus of the paper. Restructuring the text seems counterproductive. However, lines 588-598 have been added to the conclusion to address the reviewer's comment.

Point 2: The language needs to be effectively ‘beefed-up,’ it is rudimentary and somewhat inadequate.
Response 2: Extensive edits made throughout the paper, particularly focusing on diction.

Point 3: There is a tendency that the author takes knowledge of the various Korean sites referred to in the article for granted. This should be corrected so that adequate information is being provided for all sites, i.e. ref. to relevant articles or books. Korean Buddhism and its history are not widely known outside the field.
Response 3: Footnotes 4 and 49 were added and footnote 34 was edited to provide relevant background information.

Point 4: Since Hwaŏm Temple is central to the article, I would suggest that some background to this important temple be provided, esp. for the Silla period.
Response 4: Background information has been provided (see footnote 1).

Point 5: The article uses the silly Korean manner in referring to Buddhist temples by applying a double mode, i.e. ‘Hwaŏmsa Temple’, which then reads: ‘Hwaŏmsa Temple Temple.’ Silly, no?
Response 5: There are differences of opinion amongst scholars regarding the appropriate English expression of proper nouns. Though the reviewer makes a valid point, ‘Hwaŏmsa Temple’ is the official term presently used and the temple is also identified as such in the UNESCO registry. Nonetheless, the paper has been edited to eliminate the double mode from the names of all temples mentioned throughout the text (e.g., either ‘Hwaŏm temple’ or ‘Hwaŏmsa’). This also includes the temples of China and Japan referenced in the paper (e.g., either 'Ximing temple’ or ‘Ximingsi').

Point 6: In general I would discuss the use of molds and molded images within the larger context of Central and East Asian Buddhism. The practice is known from India to China and on to Korea and Japan. There are interesting, early examples from the Silk Road. At the end of the article the author extends her or his discussion to mainly Chinese examples of the Tang. That is a good thing, but it could in principle be extended even further.
Response 6: As the reviewer notes, parallel examples of molds are widely found in India, Central Asia, Tibet and Southeast Asia. However, as this paper is a study on a mold excavated from a Korean pagoda, the text focused primarily on Chinese examples of direct relevance. The author has written a separate paper titled "Extirpate Sins and Cultivate Goodness, Miniature Clay Buddha & Stupa” on the subject of molds from India and Central Asia that is in the process of publication. (See footnote 29; Reviewer makes a similar comment in 'Point 1' that is addressed in lines 588-598)

Point 7: Formally Sanskrit names should be used for Sanskrit works, and Chinese for Chinese works. Mixing them up or providing titles in Korean transcription should not be accepted for non-Korean works.
Response 7: Though the paper specifically references Chinese translations of sutras, the author provided the Sanskrit titles for texts originally written in Sanskrit. The paper has been edited as follows:
Wugoujingguang datuoluonijing (無垢淨光大陀羅尼經, Sk. Raśmivimalaviśuddhaprabhānāma-dhāraī Sūtra the Great Dharani Sūtra)’ for the first appearance in the abstract and body of the text (lines 9 & 128-129) and ‘Wugoujingguang datuoluonijing’ thereafter throughout the paper. Other titles are already properly transcribed according to their region/language of origin.

Point 8: There are a number of transcription mistakes, fx. the author mixes up two different Chinese transcription systems, eg. I-tsing when it should be Yijing. They should be corrected.
Response 8: ’I-tsing’ has been changed to 'Yijing’ in the text (see Line 292 & 295). 南海寄歸內法傳 has been edited to Nanhai jigui neifa zhuan (see line 295 & 635).

Reviewer 2 Report

The subject of this paper is a bronze Buddha mold that was used to create Buddha images from clay. The author examined various issues in relation to the bronze mold including its date of production, original function, and the reason behind its enshrinement in a pagoda. Research thus far on the function of religious art objects have focused primarily on both the visual image, such as paintings or sculptures placed in large Buddhist halls, as well as the image’s effect on the general public. This paper has a different perspective on the Buddha image as an object that allowed followers of Buddhism to maintain constant ‘remembrance’ of Buddha through ‘direct ownership’ and personal contact with a Buddha image in addition to being able to carry such an image on one’s person. The paper also recognizes the Buddha mold as a tool that made collective ritual possible. This is in part because the low cost of production was conducive to maximizing the effect and influence of the images at the least expense. Another factor is the mold’s use in the production of Buddha images was cause for the gathering of many Buddhist followers. Overall, this paper broadens the spectrum of research concerning the function of religious art. It is further meaningful in its illumination of the universal nature of all religion through the study of a specific example of Korean art. However, this is an aspect of the paper that was not addressed adequately and requires further examination in the conclusion. Moreover, it is also necessary to discuss the degree of universality in religious rituals of not only Korea but East Asia in general. Lastly, the paper does well in following the guidelines set by Religions and McCune-Reischauer is appropriately applied in the writing.
**small point
1. The caption of Figure 1 requires detailed explanation.

Author Response

RESPONSE TO REVIEWER 2

Point 1: It is further meaningful in its illumination of the universal nature of all religious through the study of a specific example of Korean art. However, this is an aspect of the paper that was not addressed adequately and requires further examination in the conclusion. Moreover, it is also necessary to discuss the degree of universality in religious rituals of not only Korea but East Asia in general.
Response 1: Lines 588-598 have been added to the conclusion to address the reviewer's comment.

Point 2: The caption of Figure 1 requires detailed explanation.
Response 2: Details added to caption of Figure 1 (see line 48).